# PROCEEDINGS A

# Research

computer modelling and simulation, applied mathematics, fluid mechanics

viscous froth model, physics of bubbles, foam rheology

**Author for correspondence:**
C. Torres-Ulloa
e-mail: carlos.torres-ulloa@strath.ac.uk

# Viscous froth model applied to the motion and topological transformations of two-dimensional bubbles in a channel: three-bubble case

## C. Torres-Ulloa and P. Grassia

Department of Chemical and Process Engineering, University of Strathclyde, James Weir Building, 75 Montrose St, Glasgow G1 1XJ, UK

CT-U, 0000-0003-4165-1339; PG, 0000-0001-5236-1850

The viscous froth model is used to predict rheological behaviour of a two-dimensional (2D) liquid-foam system. The model incorporates three physical phenomena: the viscous drag force, the pressure difference across foam films and the surface tension acting along them with curvature. In the so-called infinite staircase structure, the system does not undergo topological bubble neighbour-exchange transformations for any imposed driving back pressure. Bubbles then flow out of the channel of transport in the same order in which they entered it. By contrast, in a simple single bubble staircase or so-called lens system, topological transformations do occur for high enough imposed back pressures. The three-bubble case interpolates between the infinite staircase and simple staircase/lens. To determine at which driving pressures and at which velocities topological transformations might occur, and how the bubble areas influence their occurrence, steady-state propagating three-bubble solutions are obtained for a range of bubble sizes and imposed back pressures. As an imposed back pressure increases quasi-statically from equilibrium, complex dynamics are exhibited as the systems undergo either topological transformations, reach saddle-node bifurcation points, or asymptote to a geometrically invariant structure which ceases to change as the back pressure is further increased.

# 1. Introduction

The study of microfluidics has applications in various industries such as pharmaceuticals, medical treatment and materials formation, including metals, polymers, inorganic crystals and ceramics [1,2]. Liquid foams meanwhile have a wide range of applications, including in the mining industry, the food industry, the cosmetic industry, the production of glass, foam fractionation and firefighting [3]. Applications combining liquid-foams with microfluidics occur in processes like enhanced oil recovery (EOR) [4] and soil remediation [5], where the foam is used as a driving fluid to sweep a specific material, colloid pollutant or particles from porous media [6–8]. Using foam allows a more uniform sweep through the porous medium since foams are less sensitive to permeability heterogeneities than a Newtonian fluid would be [9]. Moreover, using foam in applications like these helps to reduce the quantity of the working fluid required, in comparison with a single-phase fluid [9]. In any of the above mentioned applications, how foam moves and rearranges inside porous media is a matter of great interest since the bubble-scale processes may affect the global foam behaviour.

The mobility of a liquid foam within porous media is affected by various factors (including liquid fraction, foam structure/bubble configuration, volume of the bubbles and the geometry of the channel of transport), all of these factors combining to make the system flow behaviour difficult to predict [10]. The complex dynamics of liquid-foams result from them seeking to reduce their interfacial energy, which is proportional to the surface area [10]. Static foams find an equilibrium state in which structures are determined via total area minimization, which leads to Plateau's laws: films connect three by three, subtending an angle of $120°$ (or $2\pi/3$) and meeting confining sidewall boundaries at an angle of $90°$ (or $\pi/2$). These constraints on film meeting angles can be considered to apply even when the foam is set into motion [11]. However, in the process of moving, the liquid interfaces or films may increase their size or else decrease their size until disappearing, leading to rearrangements of the structure [12]. Rheological complexity thereby results as a consequence of the evolution of the microstructure of the foam [13]. Since the films possess a surface tension (three-dimensional (3D) bubbles) or line tension (two-dimensional (2D) case) [14], when they deform, surface energy increases, producing stress. To compute the films' energy and the stress they produce, it is necessary to know in detail the films' positions, areas, orientations, and shapes. In fully 3D foam models, it is computationally expensive to determine energy and stress, owing to the very considerable topological and geometric complexity of the system [15]. Nevertheless, this complexity can be reduced by studying a single foam monolayer confined between two glass plates with a small separation (known as a Hele–Shaw cell). Indeed, it is possible to capture the properties of a foam layer flowing between two plates by using a 2D model known as the viscous froth model [15]. In this system, film lengths along the plates are large compared to the separation of the confining plates themselves. Viewed from above the top plate, films in the foam monolayer appear as one-dimensional (1D) curves, which is how 2D mathematical models treat them [15].

In the subsections to follow §1a–c, we review 2D foam structures, their motion and topology, which then motivates the research problem to be tackled here (described in §1d).

## (a) Foam structures in a confined system

As was demonstrated by [9], for a given driving pressure the velocity at which the liquid-foam flows through a confined plates geometry (Hele–Shaw cell), depends upon how the bubbles are arranged topologically, exhibiting discontinuities in the resulting velocities at the transition between the different topological structures such as bamboo, staircase and double staircase structures (figure 1). The reason that these different velocities result is because, depending upon how films are oriented spatially, how long they are and how fast they are moving, they experience differing amounts of viscous drag. The velocity in each structure in figure 1 is then set by the requirement that viscous drag force must be balanced by the driving pressure force. How the

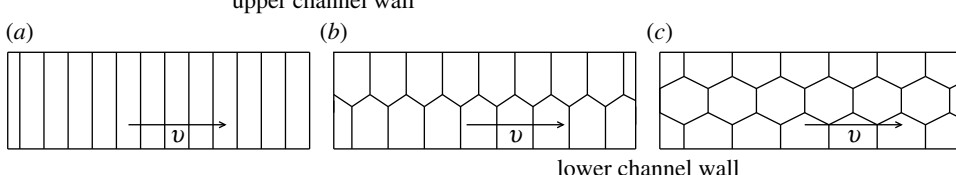

**Figure 1.** Flowing foam structures moving at velocity $v$ through a confined linear channel. Here, systems are viewed normal to the confining plates (so they appear as 2D systems) and what appear as upper and lower channel walls in this 2D view are actually sidewalls of the original 3D channel. In systems like these, films are in general 1D curves, but in all these special cases they reduce to straight lines. (a) Bamboo structure, (b) staircase structure and (c) double staircase structure.

bubbles arrange and the drag per unit velocity they thereby experience, depends on the ratio between bubble size and channel size.

The staircase structure shown in figure 1b (and likewise the double staircase in figure 1c) correspond in principle to an arbitrarily long train of bubbles moving along the channel (i.e. infinite staircases). In the case focused upon here, namely figure 1b for an arbitrarily long staircase in a straight channel and assuming monodispersity, bubbles retain the same shape no matter how far along the staircase they are nor how fast they move. Under those circumstances, for a channel of width $L$, the size of a bubble in a staircase such as figure 1b (measured from one of the channel walls to the farthest point of that bubble away from that wall) is always at least $L/2$ [13,16,17]. When this size (from channel wall to farthest point) approaches $L$, we obtain the largest monodisperse bubble area that would be permitted to stack in an infinite staircase in the fashion of figure 1b. Simple geometry gives this largest permitted area as $\sqrt{3}L^2$. Monodisperse bubbles with areas $A$ satisfying $L \leq \sqrt{A/\sqrt{3}}$ must instead select the bamboo configuration of figure 1a, although even smaller area bubbles are permitted to adopt the bamboo too. In fact, when the channel width is such that $L \leq 2\sqrt{A/\pi}$ or equivalently $\pi L^2/4 \leq A$ (with $A$ as the monodisperse bubble areas), a bamboo foam was obtained experimentally in [9]. By contrast, for $L \geq 2\sqrt{A/\pi}$ bubbles packed in a staircase or double staircase depending on the bubble size relative to channel size and operational condition of the microfluidic device along which bubbles are flowing [9,18]. Nevertheless, all above-mentioned structures, once they are originally set up, and provided they consist of arbitrarily large numbers of bubbles in a train moving along a perfectly straight channel, manage to migrate without deforming. Bubbles thereby leave the channel in the same order in which they entered it, meaning there are no bubble neighbour exchanges, or so-called $T1$ topological transformations.

What was discovered by [1] however is that when the channel is curved, such transformations become possible again. They entail that a film shrinks until it becomes zero length. Two bubbles formerly in contact then lose contact with one another, and different bubbles contact each other in their place. The precise order of occurrence of the $T1$ transformations is not known a priori [16]. Indeed whether or not they even occur at all depends upon on how rapidly the system is moving: a threshold velocity associated with a threshold imposed driving pressure is needed before they occur. Therefore, key questions of interest, in a flowing foam system, are to predict at which velocities and at which driving pressures $T1$s occur, and how the bubble areas influence their occurrence [11]. When flow is rapid, even simple cases are found to exhibit complex dynamics [12,16,17,19]. What must change between a slow moving system (no $T1$s present) and a faster moving one (with $T1$s) is the amount of viscous drag that is present. A model to predict the onset of $T1$s must therefore incorporate viscous drag in some fashion. As has been proven in [14], the viscous drag force has a nonlinear dependence with respect to the velocity $v$ (or more precisely with respect to the velocity component $v_\perp$ normal to the film). However, in this work, we consider for simplicity a linear drag law, with a drag coefficient denoted $\zeta$: this still manages to capture the key physics, i.e. $T1$ transformations only occurring beyond a threshold. We also consider a

dry foam in which film lengths greatly exceed the size of the vertices at which three films meet. Drag therefore must be assigned to film elements rather than to vertices or to film endpoints. Assumptions like these have been used in prior studies [11,13–15,17,20,21] and they lead to a simple viscous froth model that we discuss next.

## (b) Viscous froth model

The viscous froth model was originally formulated as a generalization of two situations, known as the ideal soap froth and ideal grain growth models, which can be obtained under certain limiting cases [13]. In a general flow situation, films are curved not straight, and the viscous froth model balances tension force associated with the curvature $\kappa$ of the film along the plates with the pressure difference $\Delta p$ across it, converting any mismatch between these forces into film motion, from which a viscous drag force arises [22]. The governing equation is

$$\zeta v_{\perp} = \Delta p - 2\sigma\kappa, \tag{1.1}$$

where $\Delta p$ corresponds to pressure difference across a film, measured as a back pressure minus a front pressure in the direction of motion, and $\sigma$ is surface tension, film tension being $2\sigma$. This model captures out-of-equilibrium phenomena, overcoming difficulties with previous models which produce discontinuities and jumps in film configuration if the drag term is neglected [17].

The viscous froth model has shown quantitative agreement with experiments of foam flow through curved channels [1,9,11,13]. In particular, in [1], the model was applied to a train of 12 equal-sized bubbles in the staircase structure (two bubbles across the walls as in figure 1b and several bubbles along the plates) but flowing now, not in a straight channel, but instead through a 180° bend geometry. From [1], it was demonstrated that for arbitrarily low velocities (hence arbitrarily low driving pressures), there was no $T1$ topological transformation, neither in simulation nor in experiment. On the other hand, as already mentioned, for a high flow rate (hence higher driving pressure), $T1$ topological transformations took place in the curved bend, making the foam structure unstable, both in experiment and simulation [1]. Clearly, this differs from the situation of an infinite staircase in a straight channel as described earlier.

## (c) Infinite staircase versus simple lens

The work of [1] raises the issue of whether the topological transformation observed was due to the curvature of the channel or due to the staircase having a finite number of bubbles or a mixture of both. One way to address this is to consider a finite staircase in a straight channel. In [12], the viscous froth model was applied to the motion along a straight channel of a so called lens bubble attached to one of the channel walls and a spanning film connecting the lens with the opposite channel wall: this is known as the simple lens system (figure 2a). It can be viewed as a drastic truncation of the infinite staircase in figure 1b. The lens system is (relatively) simple, but can be more susceptible to transitions than large bubble arrays because films are not surrounded by other bubbles. From [12], the lens system was found to exhibit stability when a comparatively low imposed back pressure $p_b$ is placed across it. By contrast, for higher applied pressure, the structure tends to undergo a topological transformation despite the channel being entirely straight.

The simple lens topological transformation involves a very particular route by which the structure can break up; the vertex moves upwards approaching the upper channel wall and a film that connects the vertex to that wall (front film) shrinks to zero length, and subsequently the spanning film detaches from the lens bubble, leaving this behind (figure 2b). This topological transformation involving a vertex reaching the upper wall will be called in this study a $T1_u$. This is to distinguish it from a topological transformation involving a collision between two vertices away from a wall, such as was observed in [1], which will be called here a $T1_c$: although this $T1_c$ occurred for the bubble trains in [1], it cannot occur in the simple lens system (since there is only a single vertex away from the wall, hence no other vertex with which to collide). Alternatively, a system comprised of a comparatively large lens bubble (and hence a comparatively short

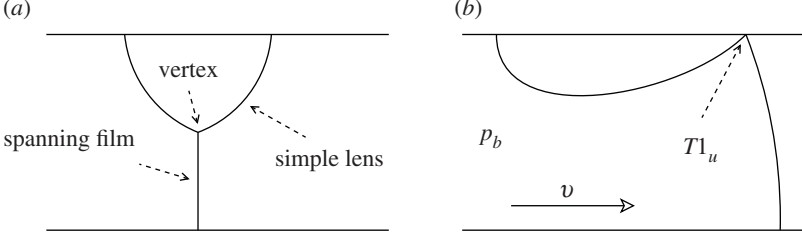

**Figure 2.** (*a*) Sketch of the simple lens problem studied in [12]. (*b*) $T1_u$ topological transformation for a structure moving at velocity $v$ due to a back pressure $p_b$ imposed upon it.

spanning film), might in principle undergo a so-called $T1_l$, where a vertex reaches the lower channel wall (to which the spanning film is connected), although this was not observed to occur in the simple lens system either [12].

If the imposed back pressure $p_b$ is increased slowly (i.e. quasi-statically, such that increases are always slow compared to the relaxation time of the structure to steady state), the simple lens system can be tracked through a sequence of different steady states. However, for the simple lens at least, the $T1_u$ is not reached quasi-statically by increasing $p_b$ from the equilibrium (by definition $p_b \equiv 0$ at equilibrium). Instead, it is reached on a second (found to be unstable in [12]) solution branch (with $p_b$ decreasing). It follows that a $T1_u$ in this case would typically be reached dynamically following loss of stability at a saddle-node bifurcation. Details are discussed further in §2e.

## (d) Finite staircase

By contrast with the simple lens but by analogy with the infinite staircase discussed earlier, it is conceivable that a truncated staircase with a *finite* number of bubbles, at least for certain choices of bubble sizes, might asymptote towards a fixed geometric structure in the limit of high imposed back pressures, without undergoing any $T1$. As per the infinite staircase then, this structure would be considered geometrically invariant. In other words, in the limit of high back pressures the geometry would cease to change, with the structure simply migrating faster and faster as back pressure increased thereafter (see figure S9 in the electronic supplementary material, §S3(a) for details of such a structure). This notwithstanding, such behaviour was never observed in the case of the simple lens, which is evidently too drastic a truncation of the infinite staircase [12]. Therefore, in order to consider the transition from $T1$s or loss of stability to geometrically invariant systems, it is necessary to explore the effect of the number of bubbles upon system behaviour. This work takes a step towards that by considering a system comprised specifically of three bubbles of various sizes arranged in a staircase structure and flowing along a confined channel. Hence, additional ways in which a staircase could break up will be explored, not just the particular mode of break up seen for the simple lens in [12]. As we will demonstrate, the mode of break up turns out to be sensitive to bubble size, or more specifically how bubble size is related to the channel width. The three-bubble system is deemed to be a next step up in complexity from the simple lens case and, as such, helps to bridge the gap between the simple lens and the infinite staircase. The main focus is to find the aforementioned topological transformations and/or saddle-node bifurcation points, for different bubble sizes, or in the absence of such situations, identify a geometrically invariant state instead. As we will find, the three-bubble system is the first situation where we see a great deal of complexity appearing in the behaviour of a staircase system (complexity at a much higher level than is present in the simple lens case in terms of how topological transformations occur). In this system, we also see the first indication that the system might start to behave qualitatively like an infinite staircase [16] i.e. exhibiting geometric invariance, albeit for the three-bubble system this is only possible in a small region

of parameter space. It is an important result though, because we also manage to prove that the simpler case, i.e. the simple lens, cannot approach a geometrically invariant state at all (see electronic supplementary material, §S3 for details). The three-bubble system studied here gives an indication of how a $N$ bubble system manages to transition from a simple lens ($N = 1$) to an infinite staircase ($N \to \infty$). An important finding though is that whether $N = 3$ behaves more akin to the simple lens or to the infinite staircase depends on the particular bubble sizes considered relative to the channel width. The methodology considered here is an entirely steady-state one, i.e. the imposed back pressure $p_b$ (or in the event that a saddle-node bifurcation is encountered as will be discussed in §2e, some other variable imposed in lieu) is varied quasi-statically, and changes in the resulting steadily propagating three-bubble structure are tracked through parameter space. This methodology is adequate to establish for which parameter sets $T1$s occur, and to classify the various types of $T1$ that are found ($T1_c$, $T1_u$ or $T1_l$ as mentioned earlier). However, an unsteady-state approach (not considered here) would be required to examine how the system evolves following any $T1$.

The rest of this work is structured as follows. In §2, we introduce a three-bubble symmetric system, symmetric in the sense that the first and third bubbles have equal size. Here, it is also shown how the structure is set up both in equilibrium and in motion. In §3, we present steady-state solution results. Conclusions are offered in §4 including the physical implication for high-speed propagation of bubbles. Additional details are relegated to electronic supplementary material, including information about the equilibrium structure (electronic supplementary material, §S1), and also methods to compute the state out of equilibrium (electronic supplementary material, §S2, with yet further details supplied in [23]). In electronic supplementary material, §S3, we obtain necessary conditions for the existence of a geometrically invariant structure out-of-equilibrium, that can migrate at arbitrarily large driving pressure. Implications of this structure are discussed in §§2–3 as pertinent. Electronic supplementary material, §S4 gives additional out-of-equilibrium results over and above those in §3.

# 2. Three-bubble symmetric system

The system studied in this work is formed of three 2D bubbles flowing through a straight channel of width $L$. In the dimensionless form of the model, as used here, $L = 1$. Across the channel width, two bubbles (bubbles $\mathcal{B}_1$ and $\mathcal{B}_3$, symmetric as they have the same size) are attached on one side of the channel (what appears as the upper channel wall in the 2D view in figure 3a) and one bubble $\mathcal{B}_2$ (possibly with a different size) is on the other side, attached to the lower channel wall. Specifically, the system is symmetric at equilibrium when both the imposed back pressure $p_b$ and the migration velocity $v$ are $p_b = v = 0$. The structure comprises seven films denoted by $J_{ij}$, where the subscript $[i, j] \in [0, 1, 2, 3]$ indicates the bubbles that each film divides (figure 3a), such that $[i, j] \equiv 0$ outside the structure. In addition, we use the superscript ‘$\circ$’ to denote variables in the equilibrium. In what follows we will consider systems both in equilibrium (figure 3a) and systems that are steadily moving, but out-of-equilibrium (figure 3b). The domain of allowed bubble areas $A_1 = A_3$ and $A_2$ in this configuration is limited (see figure S10 in electronic supplementary material, §S3 for details of the limitations).

This three-bubble structure generalizes the simple lens (figure 2), which had only a single bubble. We consider an odd number of bubbles here, since in the simple lens case interesting behaviour arose from unequal numbers of films attaching to the upper and lower walls. Like the simple lens, the three-bubble structure is also a truncation of the infinite staircase (figure 1b), albeit not quite so drastic a truncation. One of the rationales for looking at a symmetric system (i.e. bubbles $\mathcal{B}_1$ and $\mathcal{B}_3$ of the same size but not necessarily the same size as bubble $\mathcal{B}_2$) is that in a microfluidic experiment, bubbles on different sides of the channel could in principle be fed from different sources, and hence possibly have very different sizes. In the limiting case of vertex $V_2$ being close to the upper channel wall (with the length of film $J_{13}$ satisfying $l_2^{\circ} \to 0$ at the equilibrium, see figure 3a), the system might break up into two side-by-side simple lenses, as the

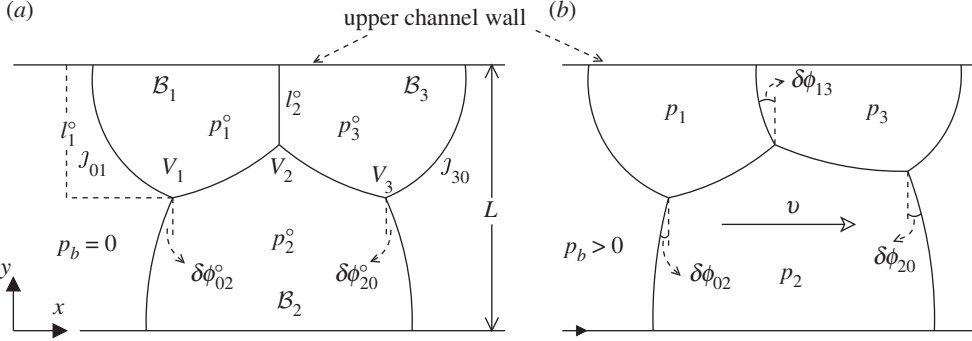

**Figure 3.** (a) Equilibrium system for channel of dimensionless width $L = 1$. The pressure of the spanning bubble $\mathcal{B}_2$ corresponds to $p_2^\circ$, the imposed back pressure to $p_b = 0$ and the pressures of the symmetric bubbles $\mathcal{B}_1$ and $\mathcal{B}_3$ are $p_1^\circ = p_3^\circ$. The distance between the upper channel wall and the vertex $V_1$ and $V_3$ is $l_1^\circ$, and between the upper channel wall and vertex $V_2$ is $l_2^\circ$. Films $J_{01}$, $J_{02}$ and $J_{12}$ join at vertex $V_1$, and films $J_{12}$, $J_{13}$ and $J_{23}$ join at vertex $V_2$, while the films $J_{23}$, $J_{30}$ and $J_{20}$ join at vertex $V_3$. Finally, the film $J_{12}$ connects vertices $V_1$ and $V_2$, and the film $J_{23}$ connects vertices $V_2$ and $V_3$. Every film forms an angle of $\pi/2$ with the respective wall of the channel and an angle $2\pi/3$ with other films. The length of the films in equilibrium are $\mathcal{L}_{01}^\circ = \mathcal{L}_{30}^\circ$, $\mathcal{L}_{12}^\circ = \mathcal{L}_{23}^\circ$ and $\mathcal{L}_{02}^\circ = \mathcal{L}_{20}^\circ$. The angles through which the films $J_{02}$ and $J_{20}$ turn are $\delta\phi_{02}^\circ = -\delta\phi_{20}^\circ$, but film $J_{13}$ is flat. (b) The system is set in motion, travelling at a unknown migration velocity $v$, as a consequence of an imposed back pressure $p_b > 0$. The film $J_{13}$ is no longer flat but turns through an unknown angle $\delta\phi_{13}$. Moreover, $\delta\phi_{02}$ and $\delta\phi_{20}$ are no longer opposite and equal.

structure is perturbed away from the equilibrium, although other modes of break up are possible also. We will return to this point in §2e(ii).

In §2a, we describe the geometry of the three-bubble system. Then, in §2b, we introduce the system's governing equations. In §2c, we characterize the equilibrium structure (further details in §S1 in electronic supplementary material). Then we characterize the steadily propagating out-of-equilibrium structure §2d (further details in §S2 in electronic supplementary material). After that in §2e, we describe the conditions to achieve a topological transformation for the three-bubble system. Finally, we introduce the notation by which the topological transformation are tracked and identified as a function of the control variable that is set to reach them (§2f). The geometry of systems that resist topological transformation altogether (i.e. that are geometrically invariant) are discussed in electronic supplementary material, §S3. Understanding all of this geometrical and topological information turns out to be relevant to the body of results that we present later on in §3.

## (a) Configuration of the three-bubble symmetric system

In the (stationary) equilibrium (figure 3a) and (steadily moving) out-of-equilibrium structure (figure 3b), the films join three by three at the respective vertices subtending an angle of $2\pi/3$ and join at an angle of $\pi/2$ with respect to the channel side walls (see also figure 4). How much a film is oriented at each point is measured with respect to the vertical in the anticlockwise direction as an angle $\phi_{ij}(s_{ij})$ (figure 3b and 4c), where $s_{ij}$ corresponds to the distance measured along a film from a wall or vertex up to a total length per film $\mathcal{L}_{ij}$, with the direction in which $s_{ij}$ is measured to be specified shortly. The orientation angle at the start of each film is expressed as $\phi_{ij}(s_{ij} = 0) \equiv \phi_{ij,0}$, and the orientation angles at the ends of films as $\phi_{ij}(s_{ij} = \mathcal{L}_{ij}) = \phi_{ij,\mathcal{L}}$. Hence, the total turning angle of each film $J_{ij}$ is then expressed as $\delta\phi_{ij} \equiv \phi_{ij,\mathcal{L}} - \phi_{ij,0}$ (with $[i,j] \in [0,1,2,3]$). In this work, we consider that for films connected with the upper channel wall, $s_{ij}$ grows downwards, where their initial orientation angle is equal to $\phi_{ij,0} \equiv 0$. Films $J_{12}$ and $J_{23}$ are also considered to have $s_{ij}$ growing downwards, or strictly speaking (given these particular films can exhibit a variety of shapes, see figure S6 in electronic supplementary material) to have $s_{ij}$ growing in the direction

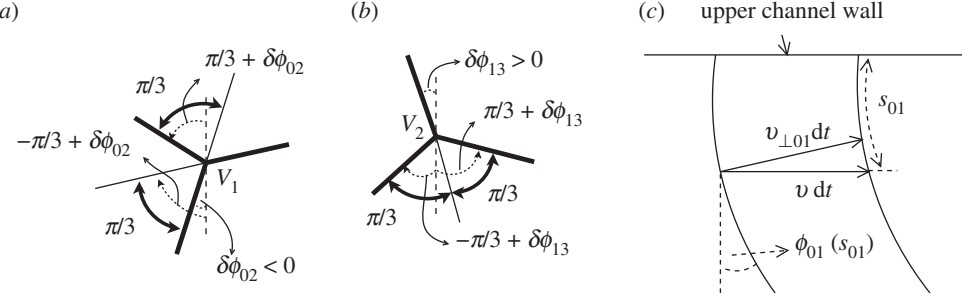

**Figure 4.** Angle measurement convention. In ($a$) and ($b$), the inset shows zoomed views near $V_1$ and $V_2$, respectively, for each film meeting at the vertex. Here, orientation angles are measured in the anticlockwise direction starting from the vertical (dashed line) to the film (thick solid line). The convention for $V_3$ (not shown) is a reflection of that for $V_1$. ($c$) View of film $J_{01}$ which is attached to the upper channel wall. At a distance $s_{01}$ measured along it, an element of the film has an orientation angle $\phi_{01}(s_{01})$ with respect to the vertical. In a time step $dt$, the element moves a distance $v_\perp\, dt$ in the normal direction and an apparent distance $v\, dt = v_{\perp 01}\, dt / \cos(\phi_{01})$ along the channel.

**Table 1.** Orientation angles $\phi_{ij}(s_{ij})$ for every film, from $s_{ij} = 0$ up to $s_{ij} = \mathcal{L}_{ij}$. Applying rules on vertex meeting angles, the initial and final orientation angles $\phi_{ij,0}$ and $\phi_{ij,\mathcal{L}}$ are expressed in terms of three (treated as independent) total turning angles $\delta\phi_{13}$, $\delta\phi_{02}$ and $\delta\phi_{20}$, respectively. More generally $\delta\phi_{ij} \equiv \phi_{ij,\mathcal{L}} - \phi_{ij,0}$.

| film | from | to | $\phi_{ij,0}$ | $\phi_{ij,\mathcal{L}}$ | $\delta\phi_{ij}$ |
|---|---|---|---|---|---|
| $J_{02}$ | lower channel wall | vertex $V_1$ | 0 | $\delta\phi_{02}$ | $\delta\phi_{02}$ |
| $J_{13}$ | upper channel wall | vertex $V_2$ | 0 | $\delta\phi_{13}$ | $\delta\phi_{13}$ |
| $J_{20}$ | lower channel wall | vertex $V_3$ | 0 | $\delta\phi_{20}$ | $\delta\phi_{20}$ |
| $J_{01}$ | upper channel wall | vertex $V_1$ | 0 | $\pi/3 + \delta\phi_{02}$ | $\pi/3 + \delta\phi_{02}$ |
| $J_{12}$ | vertex $V_2$ | vertex $V_1$ | $-\pi/3 + \delta\phi_{13}$ | $-\pi/3 + \delta\phi_{02}$ | $\delta\phi_{02} - \delta\phi_{13}$ |
| $J_{23}$ | vertex $V_2$ | vertex $V_3$ | $\pi/3 + \delta\phi_{13}$ | $\pi/3 + \delta\phi_{20}$ | $\delta\phi_{20} - \delta\phi_{13}$ |
| $J_{30}$ | upper channel wall | vertex $V_3$ | 0 | $-\pi/3 + \delta\phi_{20}$ | $-\pi/3 + \delta\phi_{20}$ |

moving away from $V_2$, which locally near $V_2$ at least is always downwards. Films $J_{12}$ and $J_{23}$ start therefore at vertex $V_2$ (with $s_{ij} = 0$), where their initial orientation angles $\phi_{12,0}$ and $\phi_{23,0}$ are expressed in terms of $\delta\phi_{13}$, as specified in table 1. By contrast, for films connected with the lower channel wall, $s_{ij}$ is considered to grow upwards, where the initial orientation angle corresponds to $\phi_{ij,0} \equiv 0$.

In this work, we specify bubble areas by fixing the vertical distance (measured down from the upper channel wall) of the vertices $V_1$ and $V_2$ in equilibrium (see §2c, and also §S1 in electronic supplementary material). These distances are denoted $l_1^\circ$ and $l_2^\circ$, respectively (figure 3a). Note that in equilibrium, vertex $V_3$ is at the same vertical location as vertex $V_1$ on symmetry grounds. Note moreover that $l_2^\circ$ is always less than $l_1^\circ$. In addition, either for equilibrium or out-of-equilibrium systems, at vertex $V_1$, we can readily express final orientation angles at the ends of films $J_{01}$ and $J_{12}$, denoted $\phi_{01,\mathcal{L}}$ and $\phi_{12,\mathcal{L}}$ respectively, in terms of $\delta\phi_{02}$. Likewise at vertex $V_3$, orientation angles for films $J_{23}$ and $J_{30}$, denoted $\phi_{23,\mathcal{L}}$ and $\phi_{30,\mathcal{L}}$, respectively, can be expressed in terms of $\delta\phi_{20}$. This is what we have summarized in table 1. As a result, only three of the total turning angles, namely $\delta\phi_{02}$, $\delta\phi_{13}$ and $\delta\phi_{20}$, are treated as being independent, the remaining turning angles $\delta\phi_{01}$, $\delta\phi_{12}$, $\delta\phi_{23}$ and $\delta\phi_{30}$ following from vertex meeting angle rules.

## (b) Model and governing equations for steady-state solution

In this section, we recall the methodology used in [12], to obtain the equations governing the steady-state film coordinates of the system. Readers familiar with this procedure from [12], may prefer to skip directly to §2c. Equation (1.1) corresponds to the dimensional form of the viscous froth model, with a linear viscous drag law. We assume typical parameter values for $L$, $\sigma$ and $\zeta$, as have been given by [12]. In this work, the viscous froth model will be used in its dimensionless form, for which spatial coordinates are rescaled by channel width $L$, bubble areas by $L^2$, $v_\perp$ is rescaled by the velocity $2\sigma/(L\zeta)$, $\Delta p$ by the pressure $2\sigma/L$, the curvature $\kappa$ by $1/L$ and finally the time scale by $L^2\zeta/(2\sigma)$ [12]. Thus the dimensionless viscous froth model applied to the motion of a local film element becomes

$$v\cos(\phi_{ij}) = v_{\perp ij} = \Delta p_{ij} - \kappa_{ij}. \tag{2.1}$$

Here, $v$ is the apparent migration velocity of the steadily propagating structure, and $\phi_{ij}$ is an orientation angle. Also, $v_{\perp ij}$ is the normal velocity, and $\Delta p_{ij}$ the pressure difference, both measured left to right. The curvature term depends on our sign convention. If $s_{ij}$ is measured downwards, we define $\kappa_{ij} = -d\phi_{ij}/ds_{ij}$; when $s_{ij}$ is measured upwards, we define instead $\kappa_{ij} = d\phi_{ij}/ds_{ij}$. With this convention, seen from downstream, convex films are always positively curved and concave films are always negatively curved. In either case, the left-hand side of equation (2.1) represents the linear viscous drag force, and the right-hand side represents the driving forces, which only balance for a static film (following Laplace's Law). Equation (2.1) is used in this work to compute the set of equations to determine film Cartesian coordinates $x_{ij}$ and $y_{ij}$ as functions of either $\phi_{ij}$ or $s_{ij}$ (see equations (S1.2)–(S1.5) for details). These coordinates need to be obtained both at equilibrium $v = 0$ (see §2c and electronic supplementary material, §S1(a)) and for out-of-equilibrium systems with $v \neq 0$ (see §2d and electronic supplementary material, §S2(a)). Based on these equations, it is possible to obtain a well specified set of system constraints (see §S2(d) in electronic supplementary material), which are then used to find steady-state solutions. This methodology was introduced in [12] for the simple lens, and how it is then adapted to model the three-bubble case is detailed in electronic supplementary material, §S2.

## (c) Equilibrium structure

For imposed back pressure $p_b = 0$, the structure is at equilibrium with $v = 0$ and $\delta\phi_{13} = 0$. Although our main interest in the present work is in moving structures with $p_b$, $v$ and $\delta\phi_{13}$ all being non-zero, understanding the equilibrium structure is important for the following reasons. Firstly, we use two equilibrium length scales $l_1^\circ$ and $l_2^\circ$ (discussed in more detail in electronic supplementary material, §S1) as surrogates for bubble areas, so it is necessary to understand how they do in fact relate to areas. Secondly varying the values of $l_1^\circ$ and $l_2^\circ$ affects all the length scales in the structure, including the lengths of all the films. Since $T1$ transformations in out-of-equilibrium structures involve films shrinking away to zero, identifying films which are already short in the equilibrium structure gives an indication of the types of $T1$ to which a system is most likely to be susceptible: more detail is given in §3a(i) and in electronic supplementary material, §S1(d).

Here as mentioned earlier equilibrium variables are denoted with the superscript '$\circ$'. The variables that define the shape of the structure are then bubble pressures $p_1^\circ$, $p_2^\circ$, $p_3^\circ$ (with $p_1^\circ = p_3^\circ$ on symmetry grounds since bubble areas $A_1$ and $A_3$ are equal), and the total turning angles $\delta\phi_{02}^\circ$ and $\delta\phi_{20}^\circ$ (with $\delta\phi_{02}^\circ = -\delta\phi_{20}^\circ$ on symmetry grounds). All these variables can be determined in terms of $l_1^\circ$ and $l_2^\circ$. At equilibrium moreover, film lengths $\mathcal{L}_{ij}^\circ$ are determined by energy minimization. Laplace's Law then applies, implying that all films except film $J_{13}$ (which is entirely flat with length $\mathcal{L}_{13}^\circ = l_2^\circ$), are arcs of circles with uniform curvature. The $x_{ij}$ and $y_{ij}$ coordinates for each film can be computed by integrating equation (2.1) for $v = 0$, as determined in §S1(a) in electronic supplementary material. At equilibrium films are arcs of circles, and bubble areas $A_1 = A_3$ and $A_2$ can then be calculated directly in terms of $p_1^\circ$, $p_2^\circ$ and $\delta\phi_{02}^\circ$, and therefore can be also computed in terms of $l_1^\circ$ and $l_2^\circ$, albeit via quite complex nonlinear equations. These are given by equations (S1.13) and (S1.14) in §S1(b) in electronic supplementary material.

## (d) Variables to capture steady-state migration

When the system is set in motion (by imposing an external force) for a given imposed back pressure $p_b$, the film coordinates $x_{ij}$ and $y_{ij}$ depend on the aforementioned turning angles, on film lengths, on bubble pressures, and on the migration velocity $v$ (figure 3b). Note that by assumption films even when moving continue to meet upper and lower channel walls at $\pi/2$ angles. This involves an assumption that the underlying Hele–Shaw system which the 2D model used here is intended to represent, has a small aspect ratio (top to bottom plate separation small relative to channel width) [15]. Details of governing equations are available in §S2(a) in electronic supplementary material. It turns out, to define a system, we have to know values of 19 variables, namely seven turning angles $\delta\phi_{ij}$ (measured from the end to the start of each film), seven film lengths $\mathcal{L}_{ij}$, three bubble pressures $p_i$, the imposed back pressure $p_b$, and migration velocity $v$. In this work, we propose two different ways of parametrizing film coordinates. The first one uses film orientation angles $\phi_{ij}$ (varying between $\phi_{ij,0}$ and $\phi_{ij,\mathcal{L}}$), and the second uses distances measured along films $s_{ij}$ (varying between 0 and $\mathcal{L}_{ij}$). If the system is parametrized in terms of $\phi_{ij}$ the number of independent variables can be reduced to 8, and if it is parametrized in terms of $s_{ij}$ to 12 (see electronic supplementary material, §S2(c) for details). The first method, having fewer variables, is simpler to implement and also closer to the method already implemented for the simple lens [12]. The second method is useful in certain systems (i.e. fast moving systems with large bubbles) in which a number of films turn out to become almost flat. Many locations on those films then have nearly the same orientation angle, but are still readily distinguished in terms of distance along the film (see electronic supplementary material, §S2, especially §S2(b), and also §S3). Using either way of parametrizing, a set of well-defined constraints must be satisfied. These correspond to three bubble area constraints (for specified film coordinates, with areas being obtained via quadrature [24]), and four (or eight) film meeting rules (films meet three by three at particular $y$ locations while subtending angles of $2\pi/3$ at vertices): we obtain four independent constraints at vertices if the system is parametrized in term of orientation angle $\phi_{ij}$, and eight if it is in terms of distances measured along films $s_{ij}$ (see electronic supplementary material, §S2(d) for details).

To summarize, depending on whether we parametrize the system in terms of orientation angles or distances measured along films, we have different numbers of independent variables and constraints to consider. However, in both cases, there is one more variable than constraint, so that one variable can be set as a control. Constraints meanwhile are determined by applying the viscous froth model to find the shape of each film, in terms of pressure difference across films $\Delta p_{ij}$ and migration velocity $v$ (see §2b). Once these film shapes are established, and film endpoints are specified (in terms of either $\delta\phi_{ij}$ or $\mathcal{L}_{ij}$), then enclosed bubble areas, vertex coordinate locations, and (if needed) orientation angles at film endpoints can all be determined (see electronic supplementary material, §S2(d) for details). The constraints are therefore merely expressed as functions (albeit complicated nonlinear functions) of the system variables. Because the constraint equations that must be solved involve nonlinear functions however, a numerical method is needed to solve them: details of the numerical method can be found in electronic supplementary material, §S2(g).

## (e) Conditions to achieve a topological transformation

Our aim here is to introduce slow quasi-static increases in imposed back pressure $p_b$ to evolve the system through a sequence of steady states that move increasingly far from equilibrium as $p_b$ increases. We anticipate however that for sufficiently high imposed back pressures a steady-state structure with the topology shown in figure 3b might not exist in all cases, so a $T1$ transformation happens. Nevertheless, for a slowly (i.e. quasi-statically) increasing back pressure $p_b$ imposed on a system, there are two conceptually distinct ways in which each of these $T1$ transformations can occur. First, a particular film might shrink quasi-statically to zero length as an imposed back pressure $p_b$ increases towards some critical pressure $p_b^*$, leading directly to $T1$. Films can then

be maintained with an arbitrarily small length for an arbitrarily long time, as long as the rate of increase of $p_b$ is low. Alternatively (as for the simple lens system in [12]; see §1c) systems can reach the end of a solution branch at a saddle-node bifurcation, such that beyond a certain critical pressure $p_b^*$ steady-state solutions cease to exist, even though all films still have a finite length at $p_b^*$. Again as for the simple lens [12], the rate of any subsequent evolution would be determined by the internal dynamics of the system, not by the rate at which an externally imposed pressure is changed. Once $p_b$ attains $p_b^*$, the internal dynamics might still drive the system to a $T1$ but on the approach to that $T1$, films can no longer be kept arbitrarily short for arbitrarily long times. How such situations are handled is described next.

### (i) Considering saddle-node bifurcations

This saddle-node scenario implies the existence of a new steady-state solution branch (typically unstable), which meets the original branch at the saddle-node bifurcation. Since $p_b$ cannot be increased beyond $p_b^*$ at steady state, in order to track this new steady solution branch away from the saddle-node bifurcation, we need to select a new control variable, usually (as in the case of [12]) one of the total turning angles $\delta\phi_{ij}$. As we track the new steady-state solution branch, the value of $p_b$ (which is now a response variable) is found to decrease. Similarly, the migration velocity $v$, and the bubble pressures $p_1$, $p_2$ and $p_3$, are expected to decrease, by an amount dependent upon the decrease in $p_b$. This was already seen in [12] for the simple lens. The expectation is that the new branch can be followed all the way to a $T1$ topological transformation, albeit with the value of $p_b$ at the $T1$ in question, now denoted $p_{b,T1}$, smaller than the aforementioned $p_b^*$. Tracking the new branch via a steady-state methodology is straightforward to do, although we cannot preclude encountering yet another saddle-node bifurcation, implying yet another solution branch to be followed, requiring yet another change of control variable. Even though the new steady solution branch itself may be dynamically unstable (hence difficult to reach from an *unsteady* state), locating and tracking it through the domain $p_b \leq p_b^*$ can still be worthwhile. By demonstrating that it joins up with the original stable solution, we prove the existence of the saddle-node bifurcation, verifying in turn that for $p_b > p_b^*$ there is no longer a corresponding steady-state solution. Mathematically speaking the saddle-node bifurcation corresponds to two branches (a stable steady solution branch and an unstable steady solution branch) meeting and annihilating [12]. However, it is important to remember also what it means in physical terms. As $p_b$ is increased, a particular edge in the structure shrinks until at a certain $p_b$ (namely $p_b^*$) the edge still has finite length, but stability of the structure is lost. Any further increase of $p_b$ beyond $p_b^*$ will result in there no longer being a steady-state structure, and so the edge in question must shrink towards a $T1$ in an unsteady fashion.

### (ii) Classifying $T1$s

Regardless of whether a $T1$ is found on an original steady solution branch or by tracking a new branch around a saddle-node bifurcation, there are several types of $T1$ of interest. A $T1_c$ happens when the film length $\mathcal{L}_{12} \to 0$ i.e. vertices $V_1$ and $V_2$ collide, and consequently $\delta\phi_{02} \to \delta\phi_{13}$ (figure 5a) or equivalently $\delta\phi_{12} \to 0$ (table 1). Meanwhile a $T1_u$ happens when $\mathcal{L}_{30} \to 0$ i.e. the vertex $V_3$ reaches the upper channel wall, and $\delta\phi_{30} \to 0$ (figure 5b) (or equivalently $\delta\phi_{20} \to \pi/3$, see table 1). When a $T1_{l1}$ takes place $\mathcal{L}_{02} \to 0$ i.e. vertex $V_1$ goes to the lower channel wall, also implying that $\delta\phi_{02} \to 0$ (figure 5c). Note however that, having $\delta\phi_{02} \to 0$ does not always imply a $T1_{l1}$, since for some given $p_b$, that film $J_{02}$ might have finite length but simply become flat between changing from being concave to convex (seen from downstream), a situation that can occur (see §S4(a) in electronic supplementary material). On the other hand, a $T1_{l3}$ takes place when $\mathcal{L}_{20} \to 0$ i.e. vertex $V_3$ goes to the lower channel wall, also implying that $\delta\phi_{20} \to 0$ (see figure 5d). A fifth scenario is also possible, as discussed in electronic supplementary material, §S1(e)(iii) and §S4(e)(iii). This corresponds to a different type of $T1_u$, denoted here as $T1_{u2}$, which takes place as $\mathcal{L}_{13} \to 0$ i.e. vertex $V_2$ goes to the upper channel wall, effectively with bubble $\mathcal{B}_2$ now cleaving the structure apart into two side-by-side simple lenses. However, as we show later

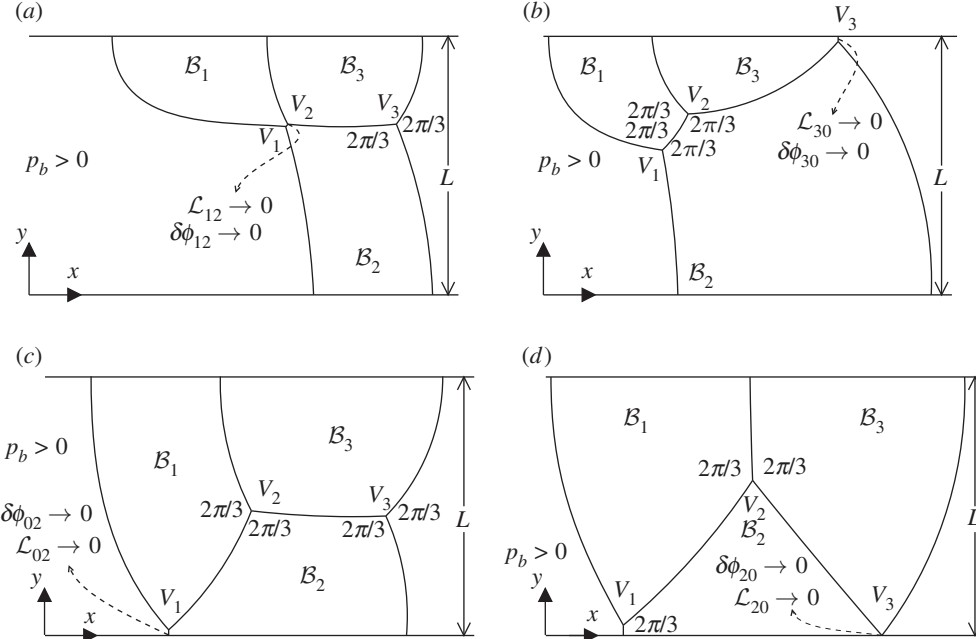

**Figure 5.** (*a*) $T1_c$ topological transformation at some $p_b > 0$ to be determined. Film length $\mathcal{L}_{12} \to 0$ while the angle $\delta\phi_{02} \to \delta\phi_{13}$, so that $\delta\phi_{12} \to 0$. (*b*) $T1_u$ topological transformation. Film length $\mathcal{L}_{30} \to 0$ while vertex $V_3$ goes to the upper channel wall. In addition $\delta\phi_{20} \to \pi/3$ so that $\delta\phi_{30} \to 0$. (*c*) $T1_{l1}$ topological transformation. Film length $\mathcal{L}_{02} \to 0$ while vertex $V_1$ goes to the lower channel wall, and $\delta\phi_{02} \to 0$. (*d*) $T1_{l3}$ topological transformation. Film length $\mathcal{L}_{20} \to 0$ while vertex $V_3$ goes to the lower channel wall, and $\delta\phi_{20} \to 0$. A fifth transformation type $T1_{u2}$ occurs only rarely and is not sketched here. It involves $V_2$ migrating to the upper channel wall, to produce two side-by-side simple lenses.

in §3a(ii), this case is just reached for systems in a very tiny domain in the limit of $l_2^\circ \ll 1$ (or equivalently $\mathcal{L}_{13}^\circ \ll 1$) and $l_1^\circ$ close to unity at the equilibrium, so therefore is of somewhat limited interest in this study. Regardless of the way in which a topological transformation occurs, as established in the numerical method to be used here, a topological transformation is considered to take place when a film length goes to $\mathcal{L}_{ij} < 10^{-6}$ (see §S2(g) in electronic supplementary material for details).

## (f) Tracking topological transformations

When film coordinates are parametrized in terms of orientation angle $\phi_{ij}$ (as in [12]; see also §S2(a) in electronic supplementary material for further details) we refer in this work to a $T1^\phi_{c,p_b}$, $T1^\phi_{u,p_b}$, $T1^\phi_{l1,p_b}$, $T1^\phi_{l3,p_b}$ and $T1^\phi_{u2,p_b}$ if the system reaches a topological transformation by quasi-static increases in $p_b$. All of the above-mentioned transformations are actually observed, and in such cases $p_b^*$ is the back pressure at which the $T1$ happens. Meanwhile $T1^\phi_{c,\delta\phi_{ij}}$, $T1^\phi_{u,\delta\phi_{ij}}$, $T1^\phi_{l1,\delta\phi_{ij}}$, $T1^\phi_{l3,\delta\phi_{ij}}$, and in principle (albeit never actually observed) $T1^\phi_{u2,\delta\phi_{ij}}$ are used to denote topological transformations found on a new solution branch, which we track following a change of control variable at a saddle-node bifurcation. Here $\delta\phi_{ij}$ is the new control variable, and typically it is chosen as the particular turning angle that is driven quasi-statically to zero as the new solution branch approaches the topological transformation. Thus we would select $\delta\phi_{12}$ on the approach to a $T1_c$, $\delta\phi_{30}$ for a $T1_u$, $\delta\phi_{02}$ for a $T1_{l1}$, $\delta\phi_{20}$ for a $T1_{l3}$, and in principle (albeit never actually observed in connection with a saddle-node bifurcation) $\delta\phi_{13}$ for a $T1_{u2}$. In any of these cases, at the $T1$ itself, generally $p_b = p_{b,T1} < p_b^*$, since $p_b^*$ corresponds now to the aforementioned saddle-node bifurcation not to the topological transformation itself. Another scenario might be found,

in which $p_b$ starts increasing again immediately before a $T1$, after first having decreased when we switched to a different control variable. Such cases are observed and will be denoted as, e.g. $T1^{\phi}_{u,\delta\phi_{30},p_b}$ if $\delta\phi_{30}$ is used a control variable, or as $T1^{\phi}_{l1,\delta\phi_{02},p_b}$ if $\delta\phi_{02}$ is used instead.

Note however (as already mentioned in §2d) we also have the option of parametrizing film coordinates by distance along films $s_{ij}$, rather than in terms of film orientation angle $\phi_{ij}$, the conversion between $s_{ij}$ and $\phi_{ij}$ being discussed in electronic supplementary material, §S2(b)). This can be convenient to do if large segments of particular films turn out to be nearly straight, meaning they have nearly the same $\phi_{ij}$ but very different $s_{ij}$. The respective topological transformations are now denoted as $T1^s_{c,p_b}$, $T1^s_{u,p_b}$, $T1^s_{l1,p_b}$, $T1^s_{l3,p_b}$ and $T1^s_{u2,p_b}$, if they are reached by quasi-static increases in $p_b$. They are denoted as $T1^s_{c,\mathcal{L}_{ij}}$, $T1^s_{u,\mathcal{L}_{ij}}$, $T1^s_{l1,\mathcal{L}_{ij}}$, $T1^s_{l3,\mathcal{L}_{ij}}$ and $T1^s_{u2,\mathcal{L}_{ij}}$ if a change of control variable onto a new solution branch is required and the topological transformation is now reached by quasi-static decreases in one of the film lengths. Here, $\mathcal{L}_{ij}$ is the specific film length that approaches zero at the topological transformation, typically $\mathcal{L}_{12}$, $\mathcal{L}_{30}$, $\mathcal{L}_{02}$, $\mathcal{L}_{20}$ or $\mathcal{L}_{13}$ in the case of $T1_c$, $T1_u$, $T1_{l1}$, $T1_{l3}$ or $T1_{u2}$, respectively. In practice, the situations we encounter here turn out to be $T1^s_{c,p_b}$ and $T1^s_{c,\mathcal{L}_{12}}$, since all the other various transformations tend to be easily reached parametrizing in terms of film orientation angle, without any need to switch to parametrize in terms of distance along films. The methodology for how we track the steady-state solution along the various solution branches up to the topological transformation is explained in §S2(f)–§S2(g) in electronic supplementary material.

In summary, in this work, we focus on steady-state systems increasing back pressure $p_b$ quasi-statically up to some critical value $p_b^*$, with some (albeit not all) cases then requiring a switch of control variable at that point, selecting either a film turning angle $\delta\phi_{ij}$ or a film length $\mathcal{L}_{ij}$ depending on how the system is parametrized. In all the above mentioned systems, topological transformations if they happen at all, are observed to occur in the following distinct ways, namely $T1_c$, $T1_u$, $T1_{l1}$, $T1_{l3}$, or (rarely) $T1_{u2}$ corresponding to vanishing of films $J_{12}$, $J_{30}$, $J_{02}$, $J_{20}$ or $J_{13}$. Which transformation occurs depends on the bubble areas $A_1 = A_3$ and $A_2$, which are defined by fixing in the equilibrium state $l_1^\circ$ and $l_2^\circ$. Nevertheless, it turns out that an alternative scenario can arise in this three-bubble system, namely that as $p_b$ is increased, a geometrically invariant structure can be reached, which does not suffer any further deformation no matter how much $p_b$ increases, such a configuration also being typical of a long train of bubbles [16]. This is described in electronic supplementary material, §S3.

# 3. Steady-state out-of-equilibrium results

In this section, we present steady-state solution results for systems driven out-of-equilibrium for a wide range of $l_1^\circ$ and $l_2^\circ$, i.e. we consider bubbles with a variety of different sizes (see tables S1 and S2 in §S1 in electronic supplementary material, to relate $l_1^\circ$ and $l_2^\circ$ to bubble areas). We find shapes of bubbles as they migrate through a confined channel, spanning the range from low to high imposed driving pressures. We start by studying the effect of having different values of $l_1^\circ$ and $l_2^\circ$ upon the different possible topological transformations that the three-bubble system might reach. This is discussed in §3a. Then, in §3b, we compare, for a selected set of systems, the maximum or critical imposed back pressure versus the pressure at which the system actually attains topological transformation. Finally, in §3c, the three-bubble system is compared with the simple lens system, by computing in each case, the imposed back pressure versus migration velocity. Additional results are relegated to electronic supplementary material. In §S4(a), we show examples in which (as in some of the cases considered in §3a(iii)) we change control variables in order to track the steady-state solution along a second solution branch. Along this particular branch, the imposed back pressure $p_b$ changes, and typically decreases as the topological transformation is approached. Information on how individual bubble pressures and system energy change as the imposed back pressure changes is given in electronic supplementary material, §S4(b) and §S4(c), respectively. In addition more information about imposed back pressures that are needed to attain topological transformation is found in electronic supplementary material, §S4(d)–§S4(e). Further

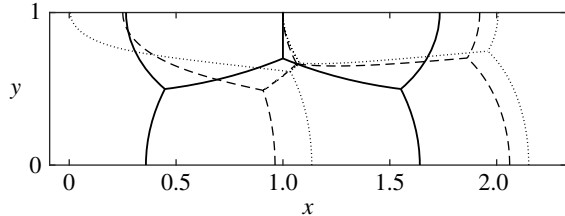

**Figure 6.** Steady-state computation for a fixed $l_1^\circ = 0.5$ and $l_2^\circ = 0.3$ in the equilibrium, and three arbitrary imposed back pressures. Solid line: $p_b = 0$. Dashed line: $p_b = 9.7037$. Dotted line: $p_b = 14.7037$. Energies (i.e. the sum of all film lengths) are, respectively, 3.5849, 3.8116 and 4.1315 so increase with $p_b$, even though some individual films shorten.

insights into the reason the system selects particular topological transformation types are offered in electronic supplementary material, §S4(f). Systems that avoid topological transformation (reaching geometrically invariant states instead) are examined in electronic supplementary material, §S4(g).

As these are steady-state computations, what matters is to establish under which conditions steadily migrating structures are admitted and for which conditions topological transformations take place (but not what happens after those transformations). The computations explore how film turning angles, film lengths, bubble pressures, and migration velocity change as the structures deform out-of-equilibrium. In figure 6, we can see, as an example, the case of a structure characterized in equilibrium by fixing $l_1^\circ = 0.5$ and $l_2^\circ = 0.3$. The resulting shape of the migrating structure is shown for three different imposed back pressures $p_b$. As the structure deforms away from equilibrium, films either grow or shorten. The majority of films grow in length, leading to an increment in the system energy (sum of lengths over all the films). However, the film lengths $\mathcal{L}_{12}$ and $\mathcal{L}_{30}$ shrink, leading in the cases in which $\mathcal{L}_{12}$ and $\mathcal{L}_{30}$ shrink away to zero to either a $T1_c$ or $T1_u$ topological transformation respectively (see discussion in §2e(ii)). In figure 6, the length of film $J_{02}$, namely $\mathcal{L}_{02}$, shrinks at first, but at higher $p_b$ grows again. The minimum of $\mathcal{L}_{02}$ turns out to happen roughly around a $p_b$ value at which the film becomes entirely flat ($\delta\phi_{02} = 0$), with the film then switching from concave to convex seen from downstream. Different combinations of $l_1^\circ$ and $l_2^\circ$ can however be found at which the length $\mathcal{L}_{02}$ or $\mathcal{L}_{20}$ shrink all the way to zero, leading to a $T1_{l1}$ or $T1_{l3}$ topological transformation, respectively.

## (a) Effect of $l_1^\circ$ versus $l_2^\circ$ upon type of topological transformation

To determine for which values of $l_1^\circ$ and $l_2^\circ$ the structure undergoes either a $T1_c$, $T1_u$, $T1_{l1}$, $T1_{l3}$ or $T1_{u2}$ topological transformation, steady-state solutions are obtained for a wide range of values of $l_1^\circ \in [0.005, 0.01, 0.015, \ldots, 0.995]$ and $l_2^\circ/l_1^\circ \in [0.005, 0.01, 0.015, \ldots, 0.995]$. For each system, we start off parametrizing films in terms of orientation angles $\phi_{ij}$, and using $p_b$ as control variable. However, the system may reach a saddle-node bifurcation at the end of a solution branch, and to track the steady-state solution to a topological transformation, we then must switch the control variable, using a procedure mentioned in §2f (see also §S2(f) in electronic supplementary material). On the other hand, if (for large $p_b$) films become very flat, it is recommended to switch from parametrizing in terms of orientation angle $\phi_{ij}$ to parametrizing in terms of distance along films $s_{ij}$ (see §S2(b) in electronic supplementary material). The domain in which we have to switch to parametrizing in terms of $s_{ij}$ is that for large values of $l_1^\circ$ and small to moderate $l_2^\circ/l_1^\circ$. By contrast, the current scheme (parametrized in terms of $\phi_{ij}$) has no issues dealing with small values of $l_1^\circ$. In the small $l_1^\circ$ regime, as we will see, systems undergo either $T1_u$ or $T1_c$ topological transformations, with $T1_u$ favoured for small $l_2^\circ/l_1^\circ$ (area $A_2$ much larger than areas $A_1 = A_3$ which are small) and $T1_c$ favoured for larger values of $l_2^\circ/l_1^\circ$ (less disparity between $A_2$ and $A_1 = A_3$ with all areas being comparatively small). This then is what we show in figures 7 and 8.

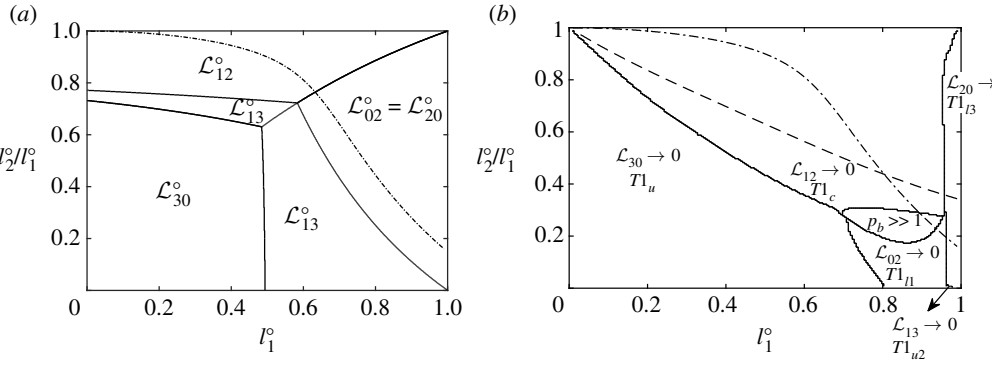

**Figure 7.** (*a*) Phase diagram dividing values of $l_1^\circ$ and $l_2^\circ/l_1^\circ$ in equilibrium into regions, each region showing which film length $\mathcal{L}_{ij}^\circ$ is the shortest one, which might make the system susceptible to undergo certain types of $T1$. Since $\mathcal{L}_{13}^\circ$ is invariably smaller than $\mathcal{L}_{30}^\circ$, the region labelled $\mathcal{L}_{30}^\circ$ is actually where $\mathcal{L}_{30}^\circ$ is *next shortest* after $\mathcal{L}_{13}^\circ$. Meanwhile the region labelled $\mathcal{L}_{13}^\circ$ is where $\mathcal{L}_{13}^\circ$ is shortest but $\mathcal{L}_{30}^\circ$ is *not* the next shortest. The dash-dotted line corresponds to the values of $l_1^\circ$ and $l_2^\circ/l_1^\circ$ for which the structure is monodisperse (all bubbles of the same size; see electronic supplementary material, §S1(c) for details). Above the dash-dotted line the area of bubbles $A_1 = A_3 > A_2$, and below it $A_1 = A_3 < A_2$. (*b*) Topological transformation phase diagram for systems set up at equilibrium with $l_1^\circ \in [0.005, 0.01, 0.015, \ldots, 0.995]$ versus $l_2^\circ/l_1^\circ \in [0.005, 0.01, 0.015, \ldots, 0.995]$, with $p_b$ then being slowly increased from the equilibrium. The different regions divided by various lines show which film length $\mathcal{L}_{ij}$ actually goes to zero, leading to $T1$ topological transformations of different types. The dash-dotted line is as per (*a*). Meanwhile in (*b*) it is only below the thin dashed line that the respective bubble areas allow the system to pack into a geometrically invariant configuration (see electronic supplementary material, §S3 for details). Such a configuration is typically seen in long trains of bubbles [16]. In the three-bubble system considered here, even though such a state can be contemplated anywhere below the thin dashed line, it is only in a small region of parameter space (labelled as $p_b \gg 1$) that the state in question is actually realized.

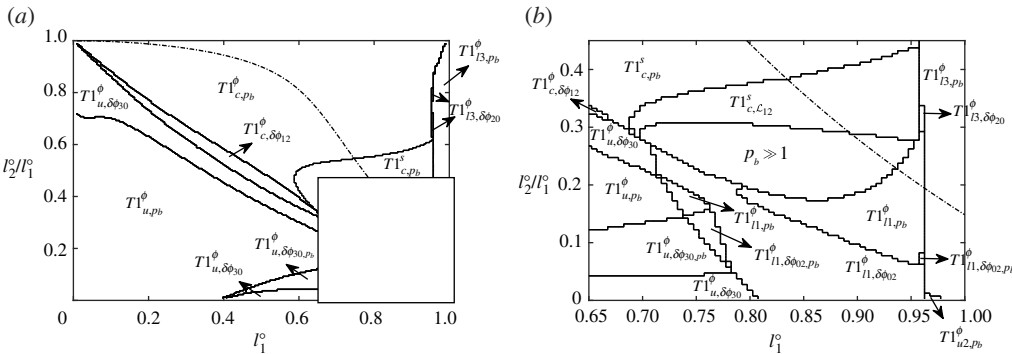

**Figure 8.** (*a*) Detailed phase diagram for different values (in equilibrium) of $l_1^\circ$ and $l_2^\circ/l_1^\circ$. Here, we show whether the system reaches a topological transformation, either a $T1_c$, $T1_u$, $T1_{l1}$, $T1_{l3}$ or a $T1_{u2}$ (see figure 5 and §2e(ii)), or else reaches the geometrically invariant migrating structure described in electronic supplementary material, §S3. Dash-dotted lines shows for which values of $l_1^\circ$ and $l_2^\circ/l_1^\circ$ the system is monodisperse. (*b*) Zoom in of (*a*).

## (i) Susceptibility to *T*1

In an effort to identify to which type of $T1$ a given system might be most susceptible, it is interesting (as mentioned in §2c) to ask whether any of the above mentioned films are already quite short *in the equilibrium state*. Accordingly, we examine the corresponding equilibrium film lengths $\mathcal{L}_{12}^\circ$, $\mathcal{L}_{30}^\circ$, $\mathcal{L}_{02}^\circ = \mathcal{L}_{20}^\circ$ and $\mathcal{L}_{13}^\circ$ for a variety of bubble sizes, i.e. a variety of $l_1^\circ$ and $l_2^\circ$. This is what figure 7*a* shows. Here, $\mathcal{L}_{13}^\circ$ is not always considered, since it is invariably shorter than $\mathcal{L}_{30}^\circ$. Provided $\mathcal{L}_{30}^\circ$ is next shortest after $\mathcal{L}_{13}^\circ$, we estimate that the system is susceptible to $T1_u$ (rather

than $T1_{u2}$) based on the fact that $T1_u$ was the mode of break up for the simple lens [12]. We only therefore consider $\mathcal{L}_{13}^\circ$ (with possible susceptibility to $T1_{u2}$) as relevant when it is shortest, but $\mathcal{L}_{30}^\circ$ is *not* the next shortest length.

What we see is that large $l_2^\circ/l_1^\circ$ shows $\mathcal{L}_{12}^\circ$ is the smallest film length (suggesting susceptibility to $T1_c$), small $l_1^\circ$ (but without very large $l_2^\circ/l_1^\circ$) shows $\mathcal{L}_{30}^\circ$ is (excluding $\mathcal{L}_{13}^\circ$) the smallest length (susceptibility to $T1_u$), and for large $l_1^\circ$ (again without very large or very small $l_2^\circ/l_1^\circ$) $\mathcal{L}_{02}^\circ = \mathcal{L}_{20}^\circ$ is the shortest length (the system might be susceptible to either $T_{l1}$ or $T_{l3}$): see §S1(d) in electronic supplementary material, for additional details on equilibrium film lengths. There is also a region of potential susceptibility to $T1_{u2}$ in which $\mathcal{L}_{13}^\circ$ is the shortest film (but $\mathcal{L}_{30}^\circ$ is *not* the next shortest). This is seen to be located mostly in the domain of large $l_1^\circ$ but very small $l_2^\circ/l_1^\circ$, but also necessarily involves a fringe immediately adjacent to the region in which $\mathcal{L}_{30}^\circ$ *is* next shortest after $\mathcal{L}_{13}^\circ$.

Although figure 7a gives an indication as to which type of $T1$ a system might be susceptible, we emphasize that this is not a definitive proof. The data in figure 7a are based entirely on the equilibrium state, and a film that starts off quite short at equilibrium, might actually grow rather than shrink as we depart from equilibrium.

### (ii) Actual type of $T1$

Figure 7b shows which type of topological transformation $T1$ the different systems *actually* undergo as a function of $l_1^\circ$ and $l_2^\circ/l_1^\circ$. Here, we show this by specifying in each case which one of the film lengths $\mathcal{L}_{ij}$ shrinks all the way down to zero, although within this figure, we make no distinction between a $T1$ reached on the original steady-state branch (reached by increasing $p_b$) and a $T1$ reached on a new branch that we switch onto after a saddle-node bifurcation. In the saddle-node case in particular, based on the findings for the simple lens [12], the original branch is typically stable while the new branch is typically unstable and we will adopt that same terminology here, despite our methodology not formally interrogating stability. The different regions on figure 7b are separated by lines. Note that these lines look slightly jagged on the plot due to sampling issues: we analysed the system for values of $l_1^\circ$ and $l_2^\circ/l_1^\circ$ selected in discrete steps of 0.005. Such jaggedness does not then reflect any underlying issue with implementing the numerical simulation technique (which is itself discussed in §S2(g)).

There is a reasonable correlation (at least in terms of how various regions are arranged with respect to one another) between the regions marked out in figure 7a (i.e. which film length $\mathcal{L}_{ij}^\circ$ is shortest) and those marked out in figure 7b (i.e. which film length $\mathcal{L}_{ij}$ ultimately vanishes at $T1$). However, the region in figure 7b in which $T1_c$ events occur (i.e. vanishing $\mathcal{L}_{12}$) is rather larger than figure 7a might suggest. Likewise the region in figure 7b in which either $T1_{l1}$ or $T1_{l3}$ events occur (i.e. vanishing $\mathcal{L}_{02}$ or $\mathcal{L}_{20}$) is rather smaller than figure 7a suggests. Additionally, there is a small region for values of $l_1^\circ$ reasonably close to unity and $l_2^\circ/l_1^\circ \ll 1$ in which the system undergoes $T1_{u2}$, i.e. vanishing $\mathcal{L}_{13}$. This $T1_{u2}$ region is *much* smaller than the region in which $\mathcal{L}_{13}^\circ$ is smallest and $\mathcal{L}_{30}^\circ$ is not the next smallest.

Monodisperse cases (dash-dotted line on figures 7a–b) tend to correspond to $T1_c$, i.e. $\mathcal{L}_{12} \to 0$, although the monodisperse line also penetrates the region in which films become exceedingly flat without $T1$ occurring (i.e. the geometrically invariant region labelled in figure 7b by $p_b \gg 1$; details in electronic supplementary material, §S3). For yet larger $l_1^\circ$, the monodisperse case also enters the region where the system undergoes $T1_{l1}$ ($\mathcal{L}_{02} \to 0$) or $T1_{l3}$ ($\mathcal{L}_{20} \to 0$).

In figure 7b, we also indicate the necessary condition derived in electronic supplementary material, §S3 for systems to admit a geometrically invariant state. This is shown by a dashed line, the entire region underneath this line meeting the necessary condition. The information presented here is the same as that in figure S10 in electronic supplementary material, §S3, just expressed in terms of $l_1^\circ$ and $l_2^\circ/l_1^\circ$ rather than in terms of $A_1$ and $A_2$. The region of parameter space within which the geometrically invariant state is actually found (labelled as $p_b \gg 1$) is significantly smaller than this. To summarize, despite the reasonable correlation between figure 7a,b, there are discrepancies between them. The reason for this is that the former figure only accounts for film lengths at

equilibrium whereas the latter considers how film lengths change away from equilibrium. This point is discussed further in electronic supplementary material, §S4(f).

### (iii) Details of $T1$ type

Figure 8 shows in detail, using the notation defined in §2f, which specific type of topological transformation the different systems undergo (if they do so), i.e. whether they approach a $T1$ via a stable or unstable solution branch, and whether the systems were parametrized in terms of $\phi_{ij}$ or $s_{ij}$ at the point of reaching it. As was the case with figure 7b, some jaggedness is evident in figure 8, but as before this is just a sampling issue, not a numerical simulation problem. Looking at figure 8a over a wide range of $l_1^\circ$ and $l_2^\circ/l_1^\circ$ values, we see that to the bottom left of the figure, systems favour $T1_{u,p_b}^\phi$, and towards the top right they favour $T1_{c,p_b}^\phi$, although on the right hand edge cases with $T1_{l3,p_b}^\phi$ are observed. Meanwhile in figure 8b (zoom in of the bottom right of figure 8a), in some cases, we see some $T1_{u2,p_b}^\phi$ transformations (albeit in a very tiny region of parameter space) and somewhat more commonly $T1_{l1,p_b}^\phi$ topological transformations. Interestingly, these $T1_{l1,p_b}^\phi$ cases appear in two distinct and disconnected lobes: a smaller lobe to the left of figure 8b and a large lobe on the right. There is a subtle difference between these lobes regarding mathematical details of how $\mathcal{L}_{12}$ approaches zero as $p_b \to p_b^*$. In both lobes, $\mathcal{L}_{12}$ falls as $p_b$ increases, but the curve of $\mathcal{L}_{12}$ versus $p_b$ might or might not exhibit an inflection point depending on the lobe.

All the above-mentioned transformations subscripted $p_b$ are on an original (believed stable) solution branch which is tracked by increasing $p_b$ monotonically. However, topological transformations $T1_{u,\delta\phi_{30}}^\phi$ and $T1_{c,\delta\phi_{12}}^\phi$ that occur on an unstable solution branch are also found for various combinations of $l_1^\circ$ and $l_2^\circ/l_1^\circ$ in figure 8a. For the most part, these tend to form 'buffer regions' between the $T1_{u,p_b}^\phi$ and $T1_{c,p_b}^\phi$ regions. In figure 8b (zoomed in), we also see examples of $T1_{l1,\delta\phi_{02}}^\phi$ and $T1_{l3,\delta\phi_{20}}^\phi$. As we have mentioned, tracking these sorts of transformations (subscripted $\delta\phi_{ij}$) involves $p_b$ reaching a maximum and decreasing again after a saddle-node bifurcation.

In certain cases, a second saddle-node bifurcation is reached when the steady-state solution is tracked in terms of $\delta\phi_{30}$ and $\delta\phi_{02}$: see regions labelled $T1_{u,\delta\phi_{30},p_b}$ and $T1_{l1,\delta\phi_{02},p_b}$. This implies that, on the final approach to the $T1$ topological change, the system has moved to yet another solution branch upon which $p_b$ starts increasing again. Thus the value of $p_b$ increases on an original solution branch, then decreases after a new control variable is selected to negotiate a saddle-node bifurcation, but finally $p_b$ starts increasing again after a second saddle-node bifurcation, immediately before the $T1$. Whether the solution branch between the second saddle-node bifurcation and the eventual $T1$ might be stable or unstable, is not a question we can interrogate with our current steady-state solution methodology.

To summarize, while saddle-node bifurcations were ubiquitous in the simple lens [12], for the three-bubble case in figure 8 they occupy a comparatively small fraction of the $l_1^\circ$ versus $l_2^\circ/l_1^\circ$ phase space. As alluded to earlier, for the most part, they form 'buffer regions' separating the various $T1_{u,p_b}^\phi$, $T1_{c,p_b}^\phi$, $T1_{l1,p_b}^\phi$ and $T1_{l3,p_b}^\phi$ regions from one another: in 'buffer regions' like these, competition between different types of $T1$ might be expected.

### (iv) Switching parametrization of the system

There are instances in which we switch from parametrizing the system in terms of orientation angle $\phi_{ij}$ to parametrizing in terms of distance along a film $s_{ij}$ (details in §2d and in electronic supplementary material, §S2(b)). The trigger for this change in parametrization is when curvature $|\kappa_{ij}|$ on at least parts of films becomes small, so that different positions on the film have nearly the same $\phi_{ij}$, albeit different $s_{ij}$. This turns out to be an issue in systems that have comparatively large bubbles and hence comparatively long films enclosing them, corresponding to cases for which $l_1^\circ$ is large and $l_2^\circ/l_1^\circ$ is small to moderate. This is exactly the region in figure 8 where the change in

parametrization occurs. What we see in figure 8a is a region of $T1^s_{c,p_b}$ topological transformations and in the zoomed view in figure 8b we see $T1^s_{c,\mathcal{L}_{12}}$ transformations also. Physically a $T1^s_{c,p_b}$ is no different from a $T1^\phi_{c,p_b}$: they both involve monotonic increases in $p_b$ up to a T1, but have merely been computed by parametrizing in different ways. Likewise, despite the different computational approach, physically there is no difference between $T1^s_{c,\mathcal{L}_{12}}$ and $T1^\phi_{c,\delta\phi_{12}}$. Both involve $p_b$ increasing up to a saddle-node bifurcation on one solution branch, and meeting a new (typically unstable) branch along which $p_b$ then decreases.

Additionally, figure 8b reveals, just as figure 7b did, that some systems also reach the geometrically invariant structure (labelled here as $p_b \gg 1$), where no further deformation in the structure can be seen (details in electronic supplementary material, §S3). A change in parametrization from $\phi_{ij}$ to $s_{ij}$ is always triggered in this case, since approaching the geometrically invariant state curvatures fall as films become asymptotically flat. Film orientations and film lengths then approach limiting values, while internal bubble pressures and migration velocity keep increasing at a constant rate as the imposed back pressure increases. We have checked (see §S4(b)–§S4(c) and §S4(g) in electronic supplementary material) that computed values of the above mentioned quantities match with predictions obtained in electronic supplementary material, §S3.

As already seen in figure 7b, just a fraction of all possible combinations of $l^\circ_1$ and $l^\circ_2/l^\circ_1$ that meet the necessary condition for geometric invariance are ultimately seen to achieve that state. In figure 8b, the domain of geometrically invariant solutions lies lower down in $l^\circ_2/l^\circ_1$ value than the region of $T1^s_{c,p_b}$ topological transformations, with the $T1^s_{c,\mathcal{L}_{12}}$ region forming a 'buffer' between the two. Other parameter regimes despite meeting the aforementioned necessary condition, e.g. those with small values of $l^\circ_1$, undergo topological transformation (typically $T1_u$) and never reach the geometrically invariant state. Systems having large $l^\circ_1$ and rather small $l^\circ_2/l^\circ_1$ exhibit, as figure 8b shows, various types of $T1_l$ transformation, e.g. $T1^\phi_{l1,p_b}$, $T1^\phi_{l1,\delta\phi_{02}}$ or $T1^\phi_{l3,p_b}$, or else (in rare cases) $T1^\phi_{u2,p_b}$, and do not reach geometrically invariant states.

In summary what we have done is to classify for which values of $l^\circ_1$ and $l^\circ_2$ systems undergo either $T1_c$, $T1_u$, $T1_{l1}$, $T1_{l3}$ or $T1_{u2}$ topological transformations (reached either quasi-statically by increasing imposed back pressure $p_b$, or else by passing through a saddle-node bifurcation), or reach the geometrically invariant migrating structure. What we have not yet discussed however are the $p_b$ values that actually must be imposed to attain these various states. This is discussed in the next section.

## (b) Critical imposed back pressure $p^*_b$ versus T1 back pressure $p_{b,T1}$

As we have already discussed, the critical imposed back pressure $p^*_b$ corresponds to the maximum allowed pressure for which a steady-state solution exists. At this point, either the system achieves a topological transformation, or else reaches the end of the current steady solution branch at a saddle-node bifurcation point. In the latter case, we can track the steady-state solution onto a second branch by using a different control variable: a turning angle $\delta\phi_{ij}$ (or one of the $\mathcal{L}_{ij}$ values), allowing us to follow what is now an unstable solution branch all the way to a topological transformation, which occurs at some $p_{b,T1}$ less than $p^*_b$. Here, we show for six fixed values of $l^\circ_1 \in [0.3, 0.5, 0.7, 0.78, 0.9, 0.97]$ and a wide range of values of $l^\circ_2/l^\circ_1 \in [0.005, 0.01, 0.015, \ldots, 0.995]$, the maximum imposed back pressure $p^*_b$ for each system, comparing these values with the pressure $p_{b,T1}$ at which the systems undergo topological transformation with $p_{b,T1} \leq p^*_b$. This is what we see in figure 9. The systems that survive out to the largest $p^*_b$ tend to be those with large $l^\circ_1$ and small to moderate $l^\circ_2/l^\circ_1$, some cases being geometrically invariant (hence having arbitrarily large $p^*_b$, within zones between vertical dashed lines in figure 9). The data in figure 9 are interesting to compare and contrast with results in [12] for the simple lens. This is done in what follows.

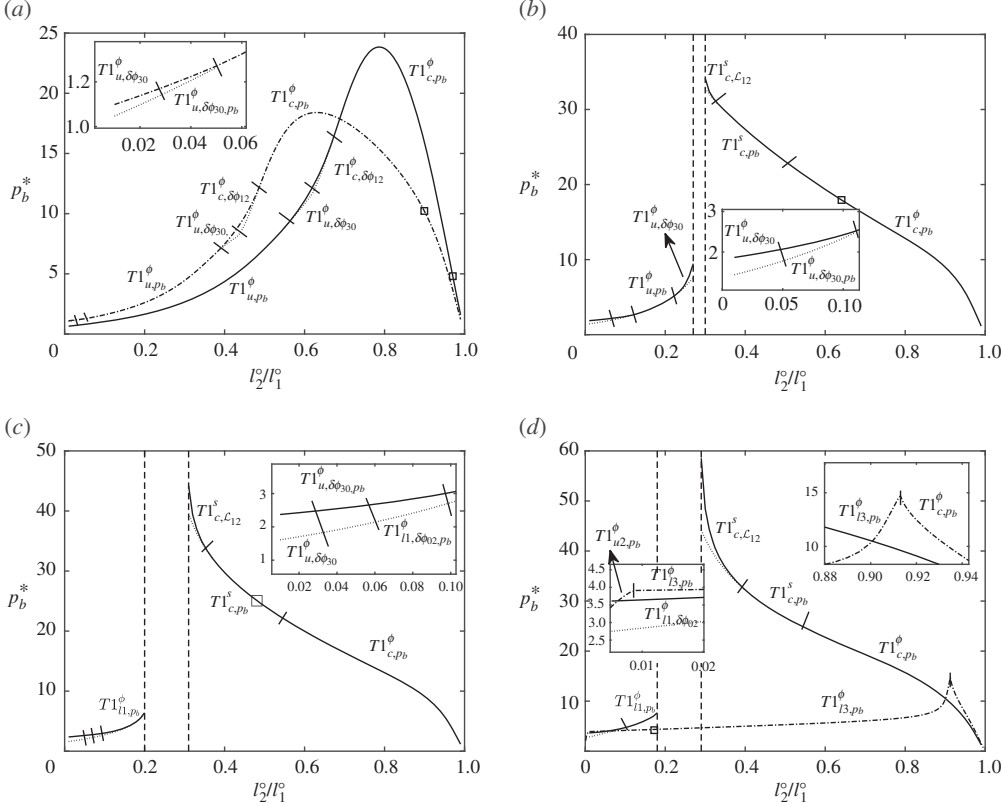

**Figure 9.** Back pressures $p_b^*$ and $p_{b,T1}$, which only differ from one another in the case of a saddle-node bifurcation (where relevant, $p_{b,T1}$ values are shown with dotted lines). The points highlighted by '□' correspond to values of $l_2^\circ/l_1^\circ$ for which systems are monodisperse. For each fixed $l_1^\circ$, the different types of $T1$, as per figure 8 are indicated. In (a), we plot data for $l_1^\circ = 0.3$ (solid line), and for $l_1^\circ = 0.5$ (dash-dotted line), in (b) for $l_1^\circ = 0.7$, in (c) for $l_1^\circ = 0.78$, and in (d) for $l_1^\circ = 0.9$ (solid line) and for $l_1^\circ = 0.97$ (dash-dotted line), in each case for $l_2^\circ/l_1^\circ \in [0.005, 0.01, 0.015, \ldots, 0.995]$. In (b)–(d), the region between the vertical dashed lines encloses systems that reach the geometrically invariant structure described in electronic supplementary material, §S3. Here values of $p_b \gg 1$ are attained.

### (i) Case of small $l_1^\circ$

When the lens bubble was small, the simple lens was found to be particularly 'rigid' or particularly 'strong', requiring a very large imposed pressure to deform significantly away from the equilibrium structure but eventually reaching a $T1_u$ topological transformation [12]. For the three-bubble system in figure 9a meanwhile, in the case $l_1^\circ = 0.3$ say (relatively small $l_1^\circ$), bubbles $\mathcal{B}_1$ and $\mathcal{B}_3$ have relatively modest area, but particularly when $l_2^\circ/l_1^\circ$ is small, the area of bubble $\mathcal{B}_2$ is much larger. Having small $l_2^\circ/l_1^\circ$ implies that film $J_{13}$ is now so short that the structure (at equilibrium at least) is almost on the point of becoming two individual simple lenses that happen to be side by side (viz. the $T1_{u2}$ transformation alluded to previously). Despite this, topological transformation is actually realized in the same fashion as for a simple lens, i.e. via $T1_u$, with film $J_{30}$ shrinking to zero. Nonetheless, as §S4(e) in electronic supplementary material explains however, the three-bubble system can (compared to the simple lens) be much more susceptible to $T1_u$ owing to the geometry of how the vertex that undergoes $T1_u$ is positioned on the bubbles. In the three-bubble system vertex $V_3$ tends, even at equilibrium, to be positioned far towards the right-hand end of bubble $\mathcal{B}_3$ meaning the bubble and vertex can easily slip apart.

Increasing $l_2^\circ/l_1^\circ$, still considering fixed $l_1^\circ = 0.3$, causes the size of bubble $\mathcal{B}_2$ to shrink. For very large $l_2^\circ/l_1^\circ$ (i.e. very close to unity) all three bubbles, $\mathcal{B}_1$, $\mathcal{B}_2$ and $\mathcal{B}_3$ are quite small area, but unlike the simple lens case, having small bubbles does not impart stability to the three-bubble structure, since the critical pressure $p_b^*$ actually decreases as $l_2^\circ/l_1^\circ$ increases (assuming $l_2^\circ/l_1^\circ$ is near unity) as figure 9a shows. The reason for having a low $p_b^*$ in this situation is that the three-bubble system undergoes a $T1_c$ topological transformation, i.e. a vertex-vertex collision away from the channel walls. There is no counterpart to this in the simple lens since, for the simple lens, there is only one single vertex away from the walls. As the electronic supplementary material, §S4(e) explains, there are ways in which a system with small $l_1^\circ$ can acquire the strength to resist $T1$ in a similar fashion to what is seen for a simple lens [12], but it requires a specific choice of $l_2^\circ/l_1^\circ$, neither too small nor too close to unity. Indeed what turns out to be crucial to governing $T1$ behaviour is exactly where on the bubble given vertices are positioned. If, on a particular bubble, they are positioned too close to neighbouring vertices and/or too close to channel walls, the system is highly susceptible to $T1$.

### (ii) Case of larger $l_1^\circ$

Now consider a much larger $l_1^\circ$ (e.g. $l_1^\circ = 0.9$ or $l_1^\circ = 0.97$ as in figure 9d) but still with comparatively large $l_2^\circ/l_1^\circ$ approaching unity. In this system, bubbles $\mathcal{B}_1$ and $\mathcal{B}_3$ are comparatively large area, but bubble $\mathcal{B}_2$ is small. Unlike the simple lens though, having this small bubble $\mathcal{B}_2$ present once again does not impart stability to the system. Again the three-bubble system undergoes topological transformation at comparatively small $p_b^*$ and again the type of transformation that occurs, namely $T1_c$, is unavailable to the simple lens.

Another contrast between the three-bubble system and the simple lens is seen for a three-bubble case in which $l_1^\circ$ is large and $l_2^\circ/l_1^\circ$ is very small. The three-bubble system then breaks up via a $T1_{u2}$, $T1_{l1}$ or $T1_{l3}$ as figure 9d shows: again this happens at a comparatively small $p_b^*$. It is noted that the $T1_{l1}$ or $T1_{l3}$ behaviour (a topological transformation at the lower channel wall), is never seen in the simple lens system, even a simple lens which at equilibrium would have a very large lens bubble connected to a very short spanning film, the latter being located near the lower channel wall. Instead, if the simple lens is deformed out of equilibrium, the spanning film lengthens significantly, and the topological transformation always occurs at the upper channel wall [12]. In a simple lens of course, the spanning film is relatively free to lengthen, since it is not associated with any bubble area constraint. The three-bubble system is however more constrained: films $J_{02}$ and $J_{20}$ connecting to the lower channel wall both contribute to an area constraint on bubble $\mathcal{B}_2$. Increasing the length of one of these (film $J_{02}$ say) might then require the length of the other (film $J_{20}$) to decrease (a requirement that close to equilibrium in the limit of small $p_b$ can actually be shown to follow on symmetry grounds), driving a $T1_{l3}$. Alternatively increasing the length of $J_{20}$ might make film $J_{02}$ shorter, leading to $T1_{l1}$. Both types of transformation are seen in figure 9d when $l_1^\circ$ is large and $l_2^\circ/l_1^\circ$ is small.

Further details of the types of topological transformations, and why certain transformations are selected for certain limiting values of $l_1^\circ$ and $l_2^\circ$ are found in electronic supplementary material, §S4(d)–§S4(e). One point discussed there is that a system with bubbles $\mathcal{B}_1$ and $\mathcal{B}_3$ small, behaves qualitatively differently (in terms of how strong it is to resist $T1$) from a system in which just bubble $\mathcal{B}_2$ is small. The difference is found to be related to the quite different bubble shapes and different film curvatures seen when $\mathcal{B}_1$ and $\mathcal{B}_3$ are small versus when $\mathcal{B}_2$ is small. One of the ways to encapsulate how the back pressure needed to drive topological transformation depends on bubbles sizes is to show a contour plot, i.e. a plot of the $l_1^\circ$ versus $l_2^\circ/l_1^\circ$ domain, with contour curves showing when the topological transformation is driven at a specified back pressure. Such plots are given in figure S20 (§S4(e)(iv) of the electronic supplementary material). These plots complement the phase diagram already given in figure 8. Whereas the phase diagram indicates the type of $T1$ transformation that occurs, the contour plots show whether the transformation happens readily or not. Having established the domain of $p_b$ over which systems preserve their topology, in the next section we examine how migration velocity $v$ behaves over that domain.

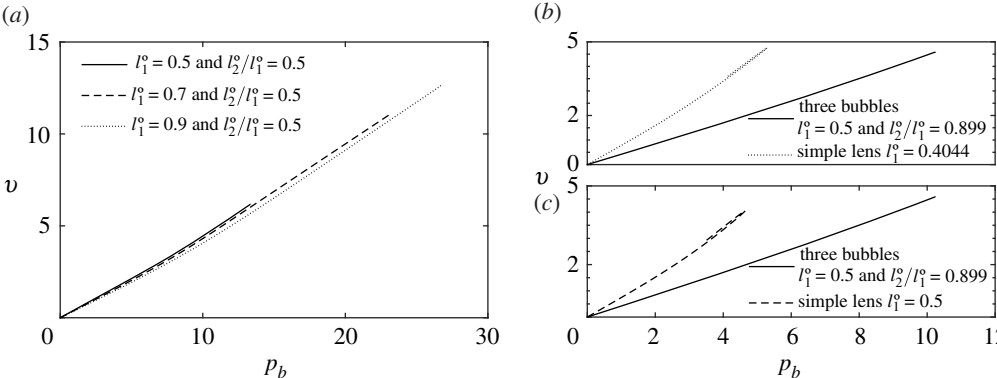

**Figure 10.** Steady-state migration velocity $v$ as a function of the imposed back pressure $p_b$. (a) Different values of $l_1^\circ \in$ [0.5, 0.7, 0.9] for the same value of $l_2^\circ/l_1^\circ = 0.5$ are considered. In (b) and (c), we plot a simple lens versus a three-bubble symmetric case system. Cases are the three-bubble system with $l_1^\circ = 0.5$ and $l_2^\circ/l_1^\circ = 0.899$ (solid line in (b) and (c)), and the simple lens with $l_1^\circ = 0.4044$ (dotted line in (b)) and with $l_1^\circ = 0.5$ (dashed line in (c)). Bubble areas are $A_1 = A_2 = A_3 = 0.1339$ for the three-bubble system. For the simple lens $A_1 = 0.1339$ ($l_1^\circ = 0.4044$) and $A_1 = 0.2047$ ($l_1^\circ = 0.5$). Note that each of the simple lens cases has two branches of $v$, one on a branch with $p_b$ increasing and the other on a branch with $p_b$ decreasing. However, the two branches almost overlay one another.

## (c) Imposed back pressure $p_b$ versus migration velocity $v$

As was shown in [12], for the simple lens, the driving velocity $v$ is a (weakly) nonlinear function of the imposed back pressure $p_b$, with smaller bubbles reaching higher critical pressures than the larger ones. The velocity was well approximated by $v \approx p_b$ for the simple lens. Here, we study for the three-bubble system how the driving velocity changes as a function of the back pressure, when different situations are considered. Specifically, we computed the driving velocity for $l_1^\circ \in$ [0.5, 0.7, 0.9], in each case with $l_2^\circ/l_1^\circ = 0.5$, up to $p_b = p_b^*$ (figure 10). We can determine that for the three-bubble system, the migration velocity is approximately $v \approx p_b/2$. This relation (which is consistent with the predictions in electronic supplementary material, §S3) comes from the fact that moving across the three-bubble structure, we must cross at least two films. For any chosen $l_1^\circ$, this relation turns out not to change significantly as we vary $l_2^\circ/l_1^\circ$, in all the studied cases, the values superposing each other on the scale of figure 10: variation of the $v$-$p_b$ relation with respect to $l_2^\circ/l_1^\circ$ is exceedingly weak. Some slight variation can be seen when different values of $l_1^\circ$ are considered (at fixed $l_2^\circ/l_1^\circ$), but even this variation is comparatively weak. Variation seen in figure 10a thereby shows the 'less weak' of two weak functions, and for simplicity in each case we plot just one fixed value of $l_2^\circ/l_1^\circ$. In figure 10a, we see that for larger values of $l_1^\circ$, the $v$ versus $p_b$ curves have very slightly lower slopes, the system travels at very slightly lower velocity.

### (i) Comparing pressures for the three-bubble system and simple lens

In figure 10b,c, we plot for $l_1^\circ = 0.5$ and $l_2^\circ/l_1^\circ = 0.899$ (a monodisperse three-bubble system) against a comparable simple lens structure obtained for the same area (figure 10b) and also a simple lens with the same $l_1^\circ = 0.5$ (figure 10c). The bubble areas in the three-bubble system are $A_1 = A_2 = A_3 = 0.1339$, whereas the bubble area in the simple lens case is $A_1 = 0.1339$ with $l_1^\circ = 0.4044$ (dotted line in figure 10b), or else is $A_1 = 0.2047$ when $l_1^\circ = 0.5$ (dashed line in figure 10c). Even the latter simple lens case here is not far from each area for the three-bubble system. There is a weak $l_1^\circ$ dependence in the $v$-$p_b$ relation in the simple lens case: larger $l_1^\circ$ gives a slightly smaller $v$ at any given $p_b$, and moreover larger $l_1^\circ$ means the system only survives out to a smaller $p_b$ (and hence a smaller $v$). These effects are predicted by [12]. In both simple lens cases, the systems reach saddle-node bifurcations: switching to a new solution branch causes migration velocity $v$

and pressure $p_b$ to start decreasing before reaching a topological transformation. However, in the simple lens cases plotted here, the 'increasing $p_b$' and 'decreasing $p_b$' solution branches have nearly the same $v$-$p_b$ relationship, namely $v \approx p_b$. Hence in each case, the data for the two branches (stable and unstable) almost overlay one another, so are only barely visible as separate branches in figure 10$b$,$c$.

This same behaviour (i.e. increasing and decreasing $p_b$ branches nearly overlaying one another) was seen for the three-bubble system in cases (albeit not plotted here) where it undergoes a saddle-node bifurcation. The present three-bubble system ($l_1^{\circ} = 0.5$, $l_2^{\circ}/l_1^{\circ} = 0.899$) however has no saddle-node bifurcation, but instead attains a $T1_c$ with $p_b$ monotonically increasing. On the other hand for the three-bubble system, the major change is that $v \approx p_b/2$ (instead of $v \approx p_b$ for the simple lens). The factor of 1/2 follows as mentioned earlier because to traverse the three-bubble structure from left to right we must cross, at the very least, two films (i.e. $J_{02}$ and $J_{20}$ which both attach to the lower channel wall).

### (ii) Comparing velocity domains for the three-bubble system and simple lens

Another observation is that the three-bubble case considered here (i.e. monodisperse case with $l_1^{\circ} = 0.5$) survives out to a higher pressure, but over almost the same velocity domain as the simple lens, regardless of whether we consider a simple lens of the same $l_1^{\circ}$ or of the same bubble area. This indicates that the particular three-bubble system considered here is at least of 'comparable strength' to the simple lens, because even though the three-bubble system survives out to higher back pressures in total (which it manages to achieve merely through having more films that must be crossed from one end of the structure to the other), it still only survives out to comparable velocities (and hence comparable imposed pressure difference per film crossed). This is potentially significant because, as we add yet more bubbles and approach the limit of an infinite staircase, those structures that eventually do exhibit topological transformation might only be stable out to a specified imposed pressure difference *per film*. On the other hand, three-bubble systems that reach a geometrically invariant configuration without topological transformation (different choices of $l_1^{\circ}$ and $l_2^{\circ}/l_1^{\circ}$ from those plotted here) survive of course out to arbitrarily large imposed pressure per film: see electronic supplementary material, §S4(d)(i) for further details of how changing $l_1^{\circ}$ impacts the relative strength of the three-bubble system and the simple lens. In summary, in the simple lens case, which consists of one bubble attached to a spanning film, the migration velocity approaches $v \approx p_b$, whereas in the three-bubble symmetric case, which consists of two bubbles of equal size plus a spanning bubble (hence two films attached to the lower channel wall), the migration velocity approaches $v \approx p_b/2$. By extension, we can deduce that for $N$ bubbles arranged in a staircase structure, the migration velocity should correspond to $v \approx 2p_b/(N+1)$. Nevertheless, we do not know definitively whether in the case when $N \gg 1$, the system always survives out to arbitrarily large $p_b$ per film, effectively reaching arbitrary large velocities $v$ also, or whether it breaks at more modest velocities. Results from the three-bubble system indicated that for certain parameter choices (i.e. certain choices of bubble areas) the structure survived out to arbitrarily large velocities, but other parameter choices only survived out to velocities comparable to those achieved in the single lens. In the three-bubble system there seems therefore to be a competition between tending to stabilize the system out to higher velocities (by adding more bubbles) versus tending to destabilize it (by allowing alternative types of topological transformations via which a structure breaks up which are not available to simpler structures). Whether, and if so how, the stabilizing tendency manages to dominate over the destabilizing one as the number $N$ of bubbles is increased even further remains an open question. Finally, note that velocity $v$ is just one response variable (albeit one readily observed in experiment) out of several that we can analyse. Other variables e.g. film turning angles, film lengths, bubble pressures and total film energies can in principle be examined, but discussion of that for the three-bubble system is relegated to electronic supplementary material, §S4(a)–§S4(c).

# 4. Conclusion

We have obtained steady-state solutions for three-bubble staircase structures (two symmetric, equal area bubbles $\mathcal{B}_1$ and $\mathcal{B}_3$ adjoining one channel wall, and $\mathcal{B}_2$, possibly of different area, adjoining the other). The structure is specified in the equilibrium by a symmetric configuration, which is set by fixing $l_1^\circ$ and $l_2^\circ$, that correspond to vertex distances from a channel wall relative to the width of the transport channel. Small values of $l_1^\circ$ represent small areas for bubbles $\mathcal{B}_1$ and $\mathcal{B}_3$, whereas large values of $l_1^\circ$ imply that bubbles $\mathcal{B}_1$ and $\mathcal{B}_3$ are larger. Moreover, for values of $l_2^\circ$ close to $l_1^\circ$ the size of bubble $\mathcal{B}_2$ is small relative to bubbles $\mathcal{B}_1$ and $\mathcal{B}_3$. By contrast, for values of $l_2^\circ \ll l_1^\circ$ the size of bubble $\mathcal{B}_2$ tends to be larger than that of $\mathcal{B}_1$ and $\mathcal{B}_3$. For any given $l_1^\circ$, a monodisperse scenario is found at some point in between these limiting cases for $l_2^\circ$.

Moving to an out-of-equilibrium state, typically by imposing a driving back pressure $p_b$ on the system, we have determined the shape of the bubbles as these systems migrate through a straight channel, looking at a range of migration velocities from low (i.e. near equilibrium) to high (large deviations from equilibrium, possibly even to the point that the structure breaks up). It is clear that the staircase structure with three bubbles, exhibits more complex and richer dynamics than the simple lens problem [12]. By tracking the steady-state solutions as $p_b$ increases, different types of topological transformations were found to cause break up of the structure. These were $T1_c$ (vertex-vertex collision), $T1_u$ (transformation at the upper wall), $T1_{l1}$ or $T1_{l3}$ (transformation at the lower wall), and $T1_{u2}$ (again transformation at the upper wall, but further upstream than $T1_u$). In the simple lens only the $T1_u$ could occur. More specifically, as the driving back pressure $p_b$ is increased slowly, possible outcomes in the three-bubble system are:

— A quasi-static $T1_c$ in which vertex $V_1$ (at the upstream end of the structure) and vertex $V_2$ (in the middle of the structure) come together and collide as the driving back pressure $p_b$ is gradually increased. This happens for a wide range of $l_1^\circ$ but with comparatively large $l_2^\circ/l_1^\circ$, hence bubble $\mathcal{B}_2$ smaller than, or of comparable size to bubbles $\mathcal{B}_1$ and $\mathcal{B}_3$.
— A quasi-static $T1_u$ in which vertex $V_3$ (at the downstream end of the structure) moves to the upper channel wall. This happens again for a range of $l_1^\circ$ but with comparatively small $l_2^\circ/l_1^\circ$, hence bubble $\mathcal{B}_2$ is rather larger than $\mathcal{B}_1$ and $\mathcal{B}_3$.
— A quasi-static $T1_{l1}$, i.e. vertex $V_1$ moves to the lower channel wall. This has large $l_1^\circ$ and very small $l_2^\circ/l_1^\circ$, hence bubbles $\mathcal{B}_1$ and $\mathcal{B}_3$ are large, while $\mathcal{B}_2$ is even slightly larger.
— A quasi-static $T1_{l3}$ i.e. vertex $V_3$ moves to the lower channel wall. This has very large $l_1^\circ$ and a range of $l_2^\circ/l_1^\circ$, hence bubbles $\mathcal{B}_1$ and $\mathcal{B}_3$ are large, but bubble $\mathcal{B}_2$ could be smaller.
— A quasi-static $T1_{u2}$ in which vertex $V_2$ moves to the upper channel wall. This happens rarely and only ever for very large $l_1^\circ$ but very small $l_2^\circ/l_1^\circ$, hence bubbles $\mathcal{B}_1$ and $\mathcal{B}_3$ are large, but bubble $\mathcal{B}_2$ is (as was the case for $T1_{l1}$) even slightly larger.
— A saddle-node bifurcation in which the aforementioned $T1_c$, $T1_u$, $T1_{l3}$ and/or $T1_{l1}$ would still occur, but they now occur dynamically rather than quasi-statically. In a phase diagram of $l_1^\circ$ versus $l_2^\circ/l_1^\circ$, these tend to form 'buffer' zones separating the various quasi-static $T1_c$, $T1_u$, $T1_{l1}$, $T1_{l3}$ and regions from one another.
— The system does not undergo any break up no matter how large the back pressure is (it reaches a geometrically invariant state). This tends to happen for large $l_1^\circ$ and small to moderate $l_2^\circ/l_1^\circ$, hence areas of all bubbles $\mathcal{B}_1$, $\mathcal{B}_2$ and $\mathcal{B}_3$ tend to be large.

Conclusions are summarized below with regard to the susceptibility of systems to various of the above mentioned transformations (§4a), what such transformations imply physically for system behaviour (§4b) and the physical implication for high-speed propagation of bubbles (§4c).

## (a) Susceptibility to different topological transformation types

Overall, highly polydisperse systems were found to be more unstable, i.e. more likely to undergo topological transformations, while monodisperse systems were found to resist them out to larger imposed pressures. In spite of the various different ways that the three-bubble structure

could break, it was found, at least in one particular case we studied, to be of comparable strength to the simple lens. It survived out to higher driving back pressures (which is expected because more films require higher pressure to move them) but it just reached similar velocities, therefore comparable imposed driving pressure per film. There were exceptions to this however, particularly when one or more bubbles were small and/or films between them were short. Full details of what happens in these small bubble and/or short film limiting cases are described in electronic supplementary material, §S4(e). The simple lens is known to be difficult to break in such cases, but the three-bubble system breaks much more readily, often via the $T1_c$ route which is not available to a simple lens. Yet another exception occurs as the aforementioned geometrically invariant state is approached: the three-bubble structure is then very difficult to break, and so survives not only out to larger pressures than the simple lens, but to larger velocities also.

Susceptibility to the different types of $T1$s and possible competition between them relies in part on the equilibrium film lengths. Broadly speaking the shortest film in the equilibrium structure gives a reasonable indication how the structure is likely to break up, and the shorter that film happens to be, the more susceptible the structure is to break up. This then explains why highly polydisperse systems tend towards instability, as typically they possess at least one film that is short at equilibrium. When out-of-equilibrium effects are taken into account however, $T1_c$ break up tends to be more common (and $T1_{l1}$, $T1_{l3}$ and $T1_{u2}$ tend to be less common) than a rule based on the shortest film length in the equilibrium structure would indicate. What is important therefore for determining $T1$-type behaviour is not just whether films are short at equilibrium but also whether they shrink or grow away from equilibrium.

## (b) Physical behaviour approaching topological transformation

For a three-bubble system, which of the various possible outcomes occurs affects not only how we go about tracking solutions mathematically, but also how the system responds physically. If the aforementioned topological transformation could be induced by tracking a single solution branch increasing imposed back pressure quasi-statically, this then implies a system could be held arbitrarily close to the transformation for an arbitrarily long time. Other cases however required tracking two distinct solution branches (a situation that was ubiquitous for the simple lens [12]).

These two distinct solution branches then meet at a saddle-node bifurcation, and were found by tracking along one branch firstly by increasing the imposed back pressure, and subsequently by varying a film turning angle, or alternatively varying a particular film length, depending how a system was parametrized. These angles and/or lengths become control variables for tracking the second steady-state solution branch, while imposed back pressure becomes a response variable and actually starts to decrease as the new steady solution branch is tracked.

As obtained for the simple lens case, here in the three-bubble system we expect that the stable solution branch is the one obtained by using imposed back pressure as control variable. The physical implication for the behaviour of the system is then as follows. As we approach the back pressure corresponding to the end of the (assumed stable) branch, all films retain finite lengths so a topological transformation has not yet occurred. Nonetheless, no steady-state solutions are permitted for any larger back pressure, so if a larger back pressure were to be imposed, the structure must evolve, presumably towards a topological transformation. As also occurred in the simple lens case, the evolution towards the transformation is expected now to be dynamic rather than quasi-static: the system can no longer be held arbitrarily close to the transformation for an indefinite period. Unsteady-state simulation (rather than the steady-state methodology used here) is then required to analyse this dynamical evolution. In fact, however, most of the three-bubble systems we have considered here allow $T1$ transformations to happen quasi-statically if the driving back pressure $p_b$ is gradually increased. Saddle-node bifurcations (with $T1$s then necessarily happening dynamically) mainly tend to occur in 'buffer zones', where two distinct edges are becoming short, so two distinct $T1$ types are then competing. This contrasts with the simple lens, for which saddle-node bifurcations were the norm rather than the exception [12].

## (c) Implications for propagating structures at high speed

Dealing with multiple solution branches and associated saddle-node bifurcations, was not the only computational challenge. For a sufficiently high imposed back pressure (hence sufficiently high speed) and for sufficiently large bubbles, films become relatively flat. It is no longer possible to compute the film coordinates in terms of a film orientation angle, since many different points on the film turn through nearly the same angle. It is then expedient to change the system coordinates, and parametrize in terms of distance measured along films instead of film orientation angle. It is only through using that parametrization, that we identified cases (with large $l_1^\circ$ and small to moderate $l_2^\circ$, i.e. with large bubbles, or more correctly large bubbles relative to channel size) that do not undergo any topological transformation whatsoever, even for an arbitrarily high imposed back pressure, suggesting the existence of a geometrically invariant state.

In such cases, the three-bubble system can therefore propagate along the channel exceedingly quickly without breaking up: a simple lens cannot do this (see electronic supplementary material, §S3). This is particularly relevant in a foam microfluidic system: if one wants to deliver a collection of bubbles (or equivalently for emulsion microfluidics, a collection of droplets) very quickly along a channel, then for a given bubble size, the channel width could be chosen as to ensure that the bubble size to channel size ratio is in the regime for which high velocities can be delivered.

As more bubbles are added to the system ($N$ bubble problem), we anticipate this geometrically invariant situation to become more common for a wider range of bubble sizes, since systems with many bubbles are expected to be able to attain arbitrarily high migration velocities (hence arbitrarily high imposed driving pressure per film), without undergoing any topological transformation. On the other hand, a structure propagating at high speed which does break up via topological transformation, might actually undergo multiple topological transformations, given that we have identified that various different types of transformation (e.g. $T1_c$, $T1_u$, $T1_{l1}$, $T1_{l3}$ and $T1_{u2}$ mentioned earlier) are now permitted. In order to determine how a system evolves after a first topological transformation and subsequently how a sequence of multiple transformations would occur, we must compute unsteady-state simulations. This will be done in future work.

What has been demonstrated without doubt here is that having three bubbles in a staircase moving in a channel can be far more complex than having just a single bubble as happens with a simple lens. This seems to echo a result from classical mechanics that it is only with three body problems that complexity suddenly appears [25]. For the case considered here of bubbles in a channel however, what remains unclear is whether having even more than three bubbles complicates the system yet further, or on the contrary somehow simplifies it through stabilizing against topological transformation. Again this question will be addressed in future work.

Another point worth emphasizing is that throughout this has been a modelling-based study. We have obtained (via the viscous froth model) predictions for the three-bubble model system regarding which types of topological transformation occur under which circumstances, but these predictions still need to be tested experimentally. Imposed pressures needed to break a structure and also pressure versus propagation velocity relationships such as have been predicted here, should in principle be possible to measure in experiment. Although a level of agreement between experiment and the viscous froth model has been reported previously in literature [1], issues still remain. One issue is that the model used here has assumed a linear relation between velocity and drag on a moving film. However, nonlinear laws can arise if the details of the film shape are sensitive to velocity [14]. Another issue is that films in this 2D model, even when moving, are assumed to meet channel walls at right angles. This then reflects that drag in the 2D model is tied to film elements and not to locations at which films terminate on channel walls. In the real Hele–Shaw system which the 2D model represents, drag is present not just along the films where they meet top and bottom plates but also on channel sidewalls. Neglecting the latter and hence neglecting any deviations away from films meeting walls at right angles requires the aspect ratio (top to bottom plate separation relative to channel width) to be small [15]. Another more general experimental issue is that foam microfluidic applications of most interest are likely to involve

many bubbles, not just three of them as considered here. In experiments then just as in modelling-based studies, the three-bubble system should be viewed primarily as a step towards generalizing to the *N*-bubble one.

Data accessibility. The data are provided in electronic supplementary material [26]. All results presented here are reproducible by analytical results and/or analytical procedures detailed in the article and the electronic supplementary material or else via numerical algorithms (accessible via https://strathcloud.sharefile.eu/d-s6a6ecaf051ea4927bbad091b29cf2475), which are also detailed in the article and the electronic supplementary material.

Authors' contributions. C.T.-U. performed mathematical analysis, developed computer code, ran computations, analysed data, prepared the first draft of the article and reviewed and edited subsequent drafts. P.G. conceived the study, provided supervision, performed mathematical analysis, analysed data and reviewed and edited article drafts. Both authors approved the final version and agree to be accountable for all aspects of the work. C.T.-U.: investigation, methodology, writing–original draft, writing–review and editing; P.G.: conceptualization, formal analysis, investigation, methodology, project administration, supervision, validation, writing–review and editing.

Competing interests. We declare we have no competing interests.

Funding. C.T.-U. received ANID Becas-Chile funding. C.T.-U. and P.G. also acknowledge support from EPSRC grant no. EP/V002937/1.

Acknowledgements. C.T.-U. acknowledges S. Cox and D. Vitasari for hosting a short research visit during which useful discussions took place, and M. Evans and S. Wilson for useful discussion during C.T.-U.'s viva voce examination.

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
