## [Peer Review File · Proceedings. Mathematical, Physical, and Engineering Sciences]

Review History

RSPA-2021-0642.R0 (Original submission)

Review form: Referee 1

Is the manuscript an original and important contribution to its field?

Excellent

Is the paper of sufficient general interest?

Good

Is the overall quality of the paper suitable?

Excellent

Can the paper be shortened without overall detriment to the main message?

No

Do you think some of the material would be more appropriate as an electronic appendix?

Yes

Do you have any ethical concerns with this paper?

No

Recommendation?

Major revision is needed (please make suggestions in comments)

Comments to the Author(s)

This is an impressive piece of work to catalogue and explain the possible outcomes when a system of three bubbles is driven along a straight channel in a Hele-Shaw cell. It has immediate applications in foam microfluidics (having more bubbles reduces the possibility of changes to the structure during flow) and possible, but perhaps currently more nebulous, applications in oil recovery and soil remediation.

The authors delineate the different bubble geometries and possible topological changes as the driving pressure slowly rises. It is undoubtedly careful painstaking work - almost 30 pages of description and a further 37 pages of supplemental material (that I didn't read carefully) to back this up - and so further discussion of the implications of the results would be welcome in the conclusions section to justify such an extensive manuscript.

Further comments on manuscript length/content:

Numerical method: the numerics are only referred to obliquely in the main body of the manuscript and delegated to the supplemental material. I think it would help the reader to understand a bit more where and why numerics are required, and how they are implemented, although I agree with the authors that the supplemental material is the right place for the details.

Film coordinates: The other topic that struck me as mysterious without having read the supplemental material is the use of film coordinates (x_{ij} , y_{ij}). I think this need to be defined in the text.

In contrast, I felt that much of section 2(c) was repetition, and by the sixth page of the results (section 3(b)) I was desperate to know what the implications are for real systems, and wonder if the rest of the results (3(b) - 3(d)) could go in the supplemental material. Section 3(c) in particular seems not to add very much.

Other than this, I have only minor comments:

Abstract: the punctuation in the abstract needs careful revision (and indeed there are many parts of the manuscript which would be easier to read with elimination/redistribution of commas and greater use of hyphens)

Section 1(c): could the authors explain what they mean by "quasi(-)statically" changing the driving pressure? Presumably that the changes are slow compared to the relaxation time of the structure to a new steady state?

Figure 2: the arrows for labels might give the impression that they indicate movement; the authors might wish to re-draw.

Section 1(d): I think it might help to emphasise that it is not necessarily the bubble size A that is important, but its relation to the channel width A/L^2 .

At the start of section 2, please note that the symmetry is only the case for $p_b = v = 0$.

Upper and lower, upwards and downwards: Perhaps it would disentangle some of the sentences (e.g. in section 1(c)) if it was made clear that "lower wall" refers to the one to which the spanning film is attached. Does s_{ij} grow "downwards" (section 2(a)) for films $_{12}$ and $_{23}$ for any p_b , or can the orientation of these films change?

Section 2(d): two methods of parametrizing the system are described, but without explaining their benefits. This only comes much later. Why not help the reader here?

Geometrically-invariant states: Is it possible to show examples of the bubble shape for these? The introduction suggests infinite staircases, as in ref [16], but that is presumably not relevant here. At the end of page 13 it is stated that these "might be reached"; is there any reason why the authors can not be definitive here?

Figure 7: the authors state that there is "a reasonable correlation" between figs 7(a) and (b). This is certainly not the case at first glance, although if by correlation the authors mean that the number of regions is the same then I might find this acceptable. I wonder if light shading could be used to indicate the correspondence that is meant? I think it would help to specify in the caption that 7(b) is the result of increasing p_b slowly from zero (if I'm correct).

Review form: Referee 2

Is the manuscript an original and important contribution to its field?

Acceptable

Is the paper of sufficient general interest?

Acceptable

Is the overall quality of the paper suitable?

Acceptable

Can the paper be shortened without overall detriment to the main message?

Yes

Do you think some of the material would be more appropriate as an electronic appendix?

Yes

Do you have any ethical concerns with this paper?

No

Recommendation?

Major revision is needed (please make suggestions in comments)

Comments to the Author(s)

Report

This paper presents simulations of a 2d soap froth in steady motion., for various cases. It uses the so-called 2D viscous froth model.

The work has been diligently compiled and presented. I have a number of reservations, which the authors might consider. If adequately implemented, they should make the paper acceptable

for publication in this journal.

(1) No experimental counterpart to these results is reported, yet the experiments are not very demanding? This is important, as the viscous froth model is clearly an idealisation, with which any given foam may or may not comply. It is always unfair in such a case to say "Go do the experiments.", but at least their prospects and the issues at stake could be addressed?

(2) Without experiments it is difficult to assess what is significant/realistic in the very detailed blow-by-blow accounts of these simulations. It would be much better if they could be presented in a much more condensed form, highlighting that main results, ie, "cutting to the chase". Many details could be relegated to appendices, supplementary material or archive.

(3) Personally I would question whether " it is possible to capture the rich properties of liquid foams by using a 2D model known as the viscous froth model" . On the face of it, this is claiming too much. The origin of the viscous effects that lie in at the heart of this model is the drag exerted by the two plates. There is no obvious counterpart in a typical 3d foam: viscous effects must arise otherwise and are likely to be very different in their effects. So I would recommend that less be claimed for the general significance of this work.

Decision letter (RSPA-2021-0642.R0)

06-Oct-2021

Dear Dr Torres Ulloa

The Editor of Proceedings A has now received comments from referees on the above paper and would like you to revise it in accordance with their suggestions which can be found below (not including confidential reports to the Editor).

Please submit a copy of your revised paper within four weeks - if we do not hear from you within this time then it will be assumed that the paper has been withdrawn. In exceptional circumstances, extensions may be possible if agreed with the Editorial Office in advance.

Please note that it is the editorial policy of Proceedings A to offer authors one round of revision in which to address changes requested by referees. If the revisions are not considered satisfactory by the Editor, then the paper will be rejected, and not considered further for publication by the journal. In the event that the author chooses not to address a referee's comments, and no scientific justification is included in their cover letter for this omission, it is at the discretion of the Editor whether to continue considering the manuscript.

To revise your manuscript, log into <http://mc.manuscriptcentral.com/prsa> and enter your Author Centre, where you will find your manuscript title listed under "Manuscripts with Decisions." Under "Actions," click on "Create a Revision." Your manuscript number has been appended to denote a revision.

You will be unable to make your revisions on the originally submitted version of the manuscript. Instead, revise your manuscript and upload a new version through your Author Centre.

When submitting your revised manuscript, you will be able to respond to the comments made by the referee(s) and upload a file "Response to Referees" in Step 1: "View and Respond to Decision Letter". Please provide a point-by-point response to the comments raised by the reviewers and

the editor(s). A thorough response to these points will help us to assess your revision quickly. You can also upload a 'tracked changes' version either as part of the 'Response to reviews' or as a 'Main document'.

IMPORTANT: Your original files are available to you when you upload your revised manuscript. Please delete any unnecessary previous files before uploading your revised version.

When revising your paper please ensure that it remains under 28 pages long. In addition, any pages over 20 will be subject to a charge (£150 + VAT (where applicable) per page). Your paper has been ESTIMATED to be 27 pages.

Open Access

You are invited to opt for open access, our author pays publishing model. Payment of open access fees will enable your article to be made freely available via the Royal Society website as soon as it is ready for publication. For more information about open access please visit <https://royalsociety.org/journals/authors/open-access/>. The open access fee for this journal is £1700/\$2380/€2040 per article. VAT will be charged where applicable. Please note that if the corresponding author is at an institution that is part of a Read and Publishing deal you are required to select this option. See <https://royalsociety.org/journals/librarians/purchasing/read-and-publish/read-publish-agreements/> for further details.

Once again, thank you for submitting your manuscript to Proc. R. Soc. A and I look forward to receiving your revision. If you have any questions at all, please do not hesitate to get in touch.

Yours sincerely
Raminder Shergill
proceedingsa@royalsociety.org

on behalf of
Professor Helen Wilson
Board Member
Proceedings A

Reviewer(s)' Comments to Author:

Referee: 1

Comments to the Author(s)

This is an impressive piece of work to catalogue and explain the possible outcomes when a system of three bubbles is driven along a straight channel in a Hele-Shaw cell. It has immediate applications in foam microfluidics (having more bubbles reduces the possibility of changes to the structure during flow) and possible, but perhaps currently more nebulous, applications in oil recovery and soil remediation.

The authors delineate the different bubble geometries and possible topological changes as the driving pressure slowly rises. It is undoubtedly careful painstaking work - almost 30 pages of description and a further 37 pages of supplemental material (that I didn't read carefully) to back this up - and so further discussion of the implications of the results would be welcome in the conclusions section to justify such an extensive manuscript.

Further comments on manuscript length/content:

Numerical method: the numerics are only referred to obliquely in the main body of the manuscript and delegated to the supplemental material. I think it would help the reader to

understand a bit more where and why numerics are required, and how they are implemented, although I agree with the authors that the supplemental material is the right place for the details.

Film coordinates: The other topic that struck me as mysterious without having read the supplemental material is the use of film coordinates (x_{ij}, y_{ij}) . I think this need to be defined in the text.

In contrast, I felt that much of section 2(c) was repetition, and by the sixth page of the results (section 3(b)) I was desperate to know what the implications are for real systems, and wonder if the rest of the results (3(b) - 3(d)) could go in the supplemental material. Section 3(c) in particular seems not to add very much.

Other than this, I have only minor comments:

Abstract: the punctuation in the abstract needs careful revision (and indeed there are many parts of the manuscript which would be easier to read with elimination/redistribution of commas and greater use of hyphens)

Section 1(c): could the authors explain what they mean by "quasi(-)statically" changing the driving pressure? Presumably that the changes are slow compared to the relaxation time of the structure to a new steady state?

Figure 2: the arrows for labels might give the impression that they indicate movement; the authors might wish to re-draw.

Section 1(d): I think it might help to emphasise that it is not necessarily the bubble size A that is important, but its relation to the channel width A/L^2 .

At the start of section 2, please note that the symmetry is only the case for $p_b = v = 0$.

Upper and lower, upwards and downwards: Perhaps it would disentangle some of the sentences (e.g. in section 1(c)) if it was made clear that "lower wall" refers to the one to which the spanning film is attached. Does s_{ij} grow "downwards" (section 2(a)) for films $_{12}$ and $_{23}$ for any p_b , or can the orientation of these films change?

Section 2(d): two methods of parametrizing the system are described, but without explaining their benefits. This only comes much later. Why not help the reader here?

Geometrically-invariant states: Is it possible to show examples of the bubble shape for these? The introduction suggests infinite staircases, as in ref [16], but that is presumably not relevant here. At the end of page 13 it is stated that these "might be reached"; is there any reason why the authors can not be definitive here?

Figure 7: the authors state that there is "a reasonable correlation" between figs 7(a) and (b). This is certainly not the case at first glance, although if by correlation the authors mean that the number of regions is the same then I might find this acceptable. I wonder if light shading could be used to indicate the correspondence that is meant? I think it would help to specify in the caption that 7(b) is the result of increasing p_b slowly from zero (if I'm correct).

Referee: 2

Comments to the Author(s)

Report

This paper presents simulations of a 2d soap froth in steady motion., for various cases. It uses the so-called 2D viscous froth model.

The work has been diligently compiled and presented. I have a number of reservations, which the authors might consider. If adequately implemented, they should make the paper acceptable for publication in this journal.

(1) No experimental counterpart to these results is reported, yet the experiments are not very demanding? This is important, as the viscous froth model is clearly an idealisation, with which any given foam may or may not comply. It is always unfair in such a case to say "Go do the experiments.", but at least their prospects and the issues at stake could be addressed?

(2) Without experiments it is difficult to assess what is significant/realistic in the very detailed blow-by-blow accounts of these simulations. It would be much better if they could be presented in a much more condensed form, highlighting that main results, ie, "cutting to the chase". Many details could be relegated to appendices, supplementary material or archive.

(3) Personally I would question whether " it is possible to capture the rich properties of liquid foams by using a 2D model known as the viscous froth model" . On the face of it, this is claiming too much. The origin of the viscous effects that lie in at the heart of this model is the drag exerted by the two plates. There is no obvious counterpart in a typical 3d foam: viscous effects must arise otherwise and are likely to be very different in their effects. So I would recommend that less be claimed for the general significance of this work.

Author's Response to Decision Letter for (RSPA-2021-0642.R0)

See Appendix A.

RSPA-2021-0642.R1 (Revision)

Review form: Referee 1

Is the manuscript an original and important contribution to its field?

Excellent

Is the paper of sufficient general interest?

Acceptable

Is the overall quality of the paper suitable?

Good

Can the paper be shortened without overall detriment to the main message?

Yes

Do you think some of the material would be more appropriate as an electronic appendix?

No

Do you have any ethical concerns with this paper?

No

Recommendation?

Accept with minor revision (please list in comments)

Comments to the Author(s)

See attached file (Appendix B).

Review form: Referee 2

Is the manuscript an original and important contribution to its field?

Good

Is the paper of sufficient general interest?

Good

Is the overall quality of the paper suitable?

Good

Can the paper be shortened without overall detriment to the main message?

Yes

Do you think some of the material would be more appropriate as an electronic appendix?

No

Do you have any ethical concerns with this paper?

No

Recommendation?

Accept as is

Comments to the Author(s)

The authors have addressed the issues raised. I now recommend publication.

Decision letter (RSPA-2021-0642.R1)

22-Dec-2021

Dear Dr Torres Ulloa,

On behalf of the Editor, I am pleased to inform you that your Manuscript RSPA-2021-0642.R1 entitled "Viscous froth model applied to the motion and topological transformations of two-dimensional bubbles in a channel: Three-bubble case" has been accepted for publication subject to minor revisions in Proceedings A. Please find the referees' comments below.

The reviewer(s) have recommended publication, but also suggest some minor revisions to your manuscript. Therefore, I invite you to respond to the reviewer(s)' comments and revise your

manuscript. Please note that we have a strict upper limit of 28 pages for each paper. Please endeavour to incorporate any revisions while keeping the paper within journal limits. Please note that page charges are made on all papers longer than 20 pages. If you cannot pay these charges you must reduce your paper to 20 pages before submitting your revision. Your paper has been ESTIMATED to be 27 pages. We cannot proceed with typesetting your paper without your agreement to meet page charges in full should the paper exceed 20 pages when typeset. If you have any questions, please do get in touch.

It is a condition of publication that you submit the revised version of your manuscript within 21 days. If you do not think you will be able to meet this date please let me know in advance of the due date.

To revise your manuscript, log into <https://mc.manuscriptcentral.com/prsa> and enter your Author Centre, where you will find your manuscript title listed under "Manuscripts with Decisions." Under "Actions," click on "Create a Revision." Your manuscript number has been appended to denote a revision.

You will be unable to make your revisions on the originally submitted version of the manuscript. Instead, revise your manuscript and upload a new version through your Author Centre.

When submitting your revised manuscript, you will be able to respond to the comments made by the referee(s) and upload a file "Response to Referees" in Step 1: "View and Respond to Decision Letter". Please provide a point-by-point response to the comments raised by the reviewers and the editor(s). A thorough response to these points will help us to assess your revision quickly. You can also upload a 'tracked changes' version either as part of the 'Response to reviews' or as a 'Main document'.

IMPORTANT: Your original files are available to you when you upload your revised manuscript. Please delete any redundant files before completing the submission process.

When uploading your revised files, please make sure that you include the following as we cannot proceed without these:

- 1) A text file of the manuscript (doc, txt, rtf or tex), including the references, tables (including captions) and figure captions. Please remove any tracked changes from the text before submission. PDF files are not an accepted format for the "Main Document".
- 2) A separate electronic file of each figure (tif, eps or print-quality pdf preferred). The format should be produced directly from original creation package, or original software format.
- 3) Electronic Supplementary Material (ESM): all supplementary materials accompanying an accepted article will be treated as in their final form. Note that the Royal Society will not edit or typeset supplementary material and it will be hosted as provided. Please ensure that the supplementary material includes the paper details where possible (authors, article title, journal name). Supplementary files will be published alongside the paper on the journal website and posted on the online figshare repository (<https://figshare.com>). The heading and legend provided for each supplementary file during the submission process will be used to create the figshare page, so please ensure these are accurate and informative so that your files can be found in searches. Files on figshare will be made available approximately one week before the accompanying article so that the supplementary material can be attributed a unique DOI. Alternatively you may upload a zip folder containing all source files for your manuscript as described above with a PDF as your "Main Document". This should be the full paper as it appears when compiled from the individual files supplied in the zip folder.

Article Funder

Please ensure you fill in the Article Funder question on page 2 to ensure the correct data is collected for FundRef (<http://www.crossref.org/fundref/>).

Media summary

Please ensure you include a short non-technical summary (up to 100 words) of the key findings/importance of your paper. This will be used for to promote your work and marketing purposes (e.g. press releases). The summary should be prepared using the following guidelines:

*Write simple English: this is intended for the general public. Please explain any essential technical terms in a short and simple manner.

*Describe (a) the study (b) its key findings and (c) its implications.

*State why this work is newsworthy, be concise and do not overstate (true 'breakthroughs' are a rarity).

*Ensure that you include valid contact details for the lead author (institutional address, email address, telephone number).

Cover images

We welcome submissions of images for possible use on the cover of Proceedings A. Images should be square in dimension and please ensure that you obtain all relevant copyright permissions before submitting the image to us. If you would like to submit an image for consideration please send your image to proceedingsa@royalsociety.org

Open Access

You are invited to opt for open access, our author pays publishing model. Payment of open access fees will enable your article to be made freely available via the Royal Society website as soon as it is ready for publication. For more information about open access please visit <https://royalsociety.org/journals/authors/open-access/>. The open access fee for this journal is £1700/\$2380/€2040 per article. VAT will be charged where applicable. Please note that if the corresponding author is at an institution that is part of a Read and Publishing deal you are required to select this option. See <https://royalsociety.org/journals/librarians/purchasing/read-and-publish/read-publish-agreements/> for further details.

Once again, thank you for submitting your manuscript to Proceedings A and I look forward to receiving your revision. If you have any questions at all, please do not hesitate to get in touch.

Best wishes

Raminder Shergill

proceedingsa@royalsociety.org

Proceedings A

Reviewer(s)' Comments to Author:

Referee: 2

Comments to the Author(s)

The authors have addressed the issues raised. I now recommend publication.

Referee: 1

Comments to the Author(s)

See attached file

Author's Response to Decision Letter for (RSPA-2021-0642.R1)

See Appendix C.

Decision letter (RSPA-2021-0642.R2)

05-Jan-2022

Dear Dr Torres Ulloa

I am pleased to inform you that your manuscript entitled "Viscous froth model applied to the motion and topological transformations of two-dimensional bubbles in a channel: Three-bubble case" has been accepted in its final form for publication in Proceedings A.

Our Production Office will be in contact with you in due course. You can expect to receive a proof of your article soon. Please contact the office to let us know if you are likely to be away from e-mail in the near future. If you do not notify us and comments are not received within 5 days of sending the proof, we may publish the paper as it stands.

As a reminder, you have provided the following 'Data accessibility statement' (if applicable). Please remember to make any data sets live prior to publication, and update any links as needed when you receive a proof to check. It is good practice to also add data sets to your reference list. Statement (if applicable): All results presented here are reproducible by analytical results and/or analytical procedures detailed in the article and the supplementary material or else via numerical algorithms (accessible via <https://strathcloud.sharefile.eu/d-s6a6ecaf051ea4927bbad091b29cf2475>), which are also detailed in the article and the supplementary material.

Under the terms of our licence to publish you may post the author generated postprint (ie. your accepted version not the final typeset version) of your manuscript at any time and this can be made freely available. Postprints can be deposited on a personal or institutional website, or a recognised server/repository. Please note however, that the reporting of postprints is subject to a media embargo, and that the status the manuscript should be made clear. Upon publication of the definitive version on the publisher's site, full details and a link should be added.

You can cite the article in advance of publication using its DOI. The DOI will take the form: 10.1098/rspa.XXXX.YYYY, where XXXX and YYYY are the last 8 digits of your manuscript number (eg. if your manuscript number is RSPA-2017-1234 the DOI would be 10.1098/rspa.2017.1234).

For tips on promoting your accepted paper see our blog post: <https://royalsociety.org/blog/2020/07/promoting-your-latest-paper-and-tracking-your-results/>

On behalf of the Editor of Proceedings A, we look forward to your continued contributions to the Journal.

Sincerely,
Raminder Shergill
proceedingsa@royalsociety.org

Appendix A

06-Oct-2021

Dear Dr Torres Ulloa

The Editor of Proceedings A has now received comments from referees on the above paper and would like you to revise it in accordance with their suggestions which can be found below (not including confidential reports to the Editor).

Please submit a copy of your revised paper within four weeks - if we do not hear from you within this time then it will be assumed that the paper has been withdrawn. In exceptional circumstances, extensions may be possible if agreed with the Editorial Office in advance.

Please note that it is the editorial policy of Proceedings A to offer authors one round of revision in which to address changes requested by referees. If the revisions are not considered satisfactory by the Editor, then the paper will be rejected, and not considered further for publication by the journal. In the event that the author chooses not to address a referee's comments, and no scientific justification is included in their cover letter for this omission, it is at the discretion of the Editor whether to continue considering the manuscript.

To revise your manuscript, log into <http://mc.manuscriptcentral.com/prsa> and enter your Author Centre, where you will find your manuscript title listed under "Manuscripts with Decisions." Under "Actions," click on "Create a Revision." Your manuscript number has been appended to denote a revision.

You will be unable to make your revisions on the originally submitted version of the manuscript. Instead, revise your manuscript and upload a new version through your Author Centre.

When submitting your revised manuscript, you will be able to respond to the comments made by the referee(s) and upload a file "Response to Referees" in Step 1: "View and Respond to Decision Letter". Please provide a point-by-point response to the comments raised by the reviewers and the editor(s). A thorough response to these points will help us to assess your revision quickly. You can also upload a 'tracked changes' version either as part of the 'Response to reviews' or as a 'Main document'.

IMPORTANT: Your original files are available to you when you upload your revised manuscript. Please delete any unnecessary previous files before uploading your revised version.

When revising your paper please ensure that it remains under 28 pages long. In addition, any pages over 20 will be subject to a charge (£150 + VAT (where applicable) per page). Your paper has been ESTIMATED to be 27 pages.

Once again, thank you for submitting your manuscript to Proc. R. Soc. A and I look forward to receiving your revision. If you have any questions at all, please do not hesitate to get in touch.

Yours sincerely

Raminder Shergill
proceedingsa@royalsociety.org

on behalf of
Professor Helen Wilson
Board Member
Proceedings A

15-Oct-2021

Dear Editor,

Thank you for sending the reviewer reports on the manuscript ID RSPA-2021-0642. We have addressed all the concerns of the reviewers, making the corresponding changes to the documents (main text and supplementary material), and are resubmitting a new version of the manuscript.

As with any submission, we confirm that this manuscript has not been previously published, nor is it under consideration for publication by any other journal. Both authors are aware of the manuscript and approve its submission and agree to be accountable for all aspects of the work. For the convenience of the editor and reviewers, changes within the main text and supplementary material are highlighted in colour.

We also confirm that we are aware of the article length restrictions applicable to Proc. Roy. Soc. A. In the event that our manuscript is accepted for publication and attracts page charges, we confirm that we agree to meet those charges.

To address the concerns of reviewer 1, we have reduced the extension of the results section by moving section 3(c) to the supplementary material. Also, in an effort to avoid repetition in section 2(c), this has been reduced in length. In addition, as reviewer 2 also pointed out, further discussion on the experimental counterpart related to viscous froth model has been included in the introduction section, and more details about the possible experimental implications of the results have also been included in the conclusion section. We have also provided a detailed point-by-point response to follow.

Yours sincerely,

Carlos Torres-Ulloa and Paul Grassia

Reviewer(s)' Comments to Author:

Referee: 1

Comments to the Author(s)

This is an impressive piece of work to catalogue and explain the possible outcomes when a system of three bubbles is driven along a straight channel in a Hele-Shaw cell. It has immediate applications in foam microfluidics (having more bubbles reduces the possibility of changes to the structure during flow) and possible, but perhaps currently more nebulous, applications in oil recovery and soil remediation.

The authors delineate the different bubble geometries and possible topological changes as the driving pressure slowly rises. It is undoubtedly careful painstaking work - almost 30 pages of description and a further 37 pages of supplemental material (that I didn't read carefully) to back this up - and so further discussion of the implications of the results would be welcome in the conclusions section to justify such an extensive manuscript.

This seems to echo reviewer 2, who mentioned possible issues at stake for experiments as being a point that needed to be mentioned. In our response to reviewer 2 we included a discussion of experimental outlook right at the end of the conclusions, and we feel this may also satisfy the point reviewer 1 is making here.

Further comments on manuscript length/content:

Numerical method: the numerics are only referred to obliquely in the main body of the manuscript and delegated to the supplemental material. I think it would help the reader to understand a bit more where and why numerics are required, and how they are implemented, although I agree with the authors that the supplemental material is the right place for the details.

The reason why numerics are needed has been explained. Specifically a brief explanation was included in section 2(d).

Film coordinates: The other topic that struck me as mysterious without having read the supplemental material is the use of film coordinates (x_{ij}, y_{ij}) . I think this need to be defined in the text.

This notation has been defined now in section 2(b).

In contrast, I felt that much of section 2(c) was repetition, and by the sixth page of the results (section 3(b)) I was desperate to know what the implications are for real systems, and wonder if the rest of the results (3(b) - 3(d)) could go in the supplemental material. Section 3(c) in particular seems not to add very much.

Section 2(c) has been reduced in length. There were indeed some sentences that were repetitive in section 2(c) and these have been eliminated. Former section 3(c) has been moved to supplementary material, now as section S 4(a). We argue for retaining sections 3(b) and former section 3(d) (now section 3(c)) in main text as they are sections that correspond to measurements an experimentalist would be likely to make (topological break up vs driving pressure; migration velocity vs driving pressure); this is a point we now make in the final paragraph of 3(c)(ii) and also in the final paragraph of the conclusions. Meanwhile the sorts of quantities discussed in former section 3(c) (now supplementary section S 4(a)), namely film turning angles and film lengths, could in principle be measured experimentally e.g. using image processing techniques, but these are arguably not the first measurements an experimentalist would choose to make. Hence we agree this section should be supplementary material.

Other than this, I have only minor comments:

Abstract: the punctuation in the abstract needs careful revision (and indeed there are many parts of the manuscript which would be easier to read with elimination/redistribution of commas and greater use of hyphens)

The punctuation has been checked throughout. Early on in the abstract, there was a clumsy sentence with far too many commas. We broke this up into two sentences to make it clearer.

Section 1(c): could the authors explain what they mean by “quasi(-)statically” changing the driving pressure? Presumably that the changes are slow compared to the relaxation time of the structure to a new steady state?

This does indeed need to be explained, and the reviewer’s explanation is correct. In the quasistatic limit, changes in p_b are arbitrarily slow compared to the relaxation time of the structure to steady state, so the system effectively evolves through a sequence of steady states. Section 1(c) is modified accordingly.

Figure 2: the arrows for labels might give the impression that they indicate movement; the authors might wish to re-draw.

Figure 2 (and also Figure 3) has been changed as suggested.

Section 1(d): I think it might help to emphasise that it is not necessarily the bubble size A that is important, but its relation to the channel width A/L^2 .

It has now been mentioned in section 1(d) that what matters is bubble size relative to channel size.

At the start of section 2, please note that the symmetry is only the case for $p_b = v = 0$.

This has been stated explicitly at the start of section 2.

Upper and lower, upwards and downwards: Perhaps it would disentangle some of the sentences (e.g. in section 1(c)) if it was made clear that “lower wall” refers to the one to which the spanning film is attached. Does s_{ij} grow “downwards” (section 2(a)) for films j_{12} and j_{23} for any p_b , or can the orientation of these films change?

The point that the lower wall is the wall to which the spanning film connects has been made clear in section 1(c). In general we have tried to use terminology consistently throughout. We use the term “upper wall” and “lower wall” to refer to the 2D view, whereas we use “top plate” (but never “upper plate”) (in section 1) to refer to a 3D view. On the other hand, in section 2(a), “downwards” in this sense for films j_{12} and j_{23} means “moving away from V_2 ”. The reviewer is correct that there are cases (in Figure S 6 in the supplementary material for instance) in which these films start moving downwards away from vertex V_2 and then swing back upwards to reach vertex V_1 and V_3 , although V_1 and V_3 still end up lower down than V_2 started, so the net motion is still downwards. For avoidance of doubt though we have reworded section 2(a).

Section 2(d): two methods of parametrizing the system are described, but without explaining their benefits. This only comes much later. Why not help the reader here?

This has been mentioned in section 2(d).

Geometrically-invariant states: Is it possible to show examples of the bubble shape for these? The introduction suggests infinite staircases, as in ref [16], but that is presumably not relevant here. At the end of page 13 it is stated that these “might be reached”; is there any reason why the authors can not be definitive here?

In section 1(d) we have reworded to emphasise that in the three-bubble system (unlike the infinite staircase), the structure is not invariant over the full range of back pressures, only in high pressure limit. Also in the first paragraph in section 1(d) the reader is pointed to a figure showing the geometrically invariant state (Figure S 9 in the supplementary material). In addition, “might be reached” has been changed to “can be reached” (in section 2(f)). The original wording “might” was intended to reflect the notion that there is a necessary condition (in terms of the parameters l_1^o and l_2^o) for existence of the geometrically invariant state, but not all parameter sets meeting that necessary condition attained the state. It is a necessary condition after all, but not sufficient. On balance though we agree with the reviewer that the word “might” is a little distracting in this context, so have changed it.

Figure 7: the authors state that there is “a reasonable correlation” between figs 7(a) and (b). This is certainly not the case at first glance, although if by correlation the authors mean that the number of regions is the same then I might find this acceptable.

This has been reworded (in section 3(a)(ii)). It is not just the number of regions that is the same, there is also a correlation in terms of where they are located with respect to each other in the diagram. What differs of course between the diagrams is the size of the regions as section 3(a)(ii) goes on to explain.

I wonder if light shading could be used to indicate the correspondence that is meant? I think it would help to specify in the caption that 7(b) is the result of increasing p_b slowly from zero (if I’m correct).

The explanation has been given in words (as alluded to above); as mentioned, the correspondence here concerns the spatial arrangement of regions, albeit not their size. We feel shading would be very distracting (particularly when we move from Figure 7(b) to Figure 8 which has a multitude of regions).

Referee: 2

Comments to the Author(s)

This paper presents simulations of a 2d soap froth in steady motion., for various cases. It uses the so-called 2D viscous froth model.

The work has been diligently compiled and presented. I have a number of reservations, which the authors might consider. If adequately implemented, they should make the paper acceptable for publication in this journal.

(1) No experimental counterpart to these results is reported, yet the experiments are not very demanding? This is important, as the viscous froth model is clearly an idealisation, with which any given foam may or may not comply. It is always unfair in such a case to say “Go do the experiments.”, but at least their prospects and the issues at stake could be addressed?

This is a valid point and discussion has been added, in the introduction (section 1), in section 2 (see the second paragraph of section 2), in section 3(c)(ii) (see the last paragraph of that section), and in the conclusions (see the last paragraph of the conclusions). Note that a number of the claims we made in section 1 were actually describing experimental systems (not just simulation studies), but it was left implicit: it is now made explicit. One general comment we would make however is that doing an experiment to test (and possibly disprove) a specific set of predictions is better than doing an experiment blindly without any predictions available to test. The intellectual challenge in this manuscript has been to obtain the predictions. An experimental study will no doubt present a different set of intellectual challenges.

(2) Without experiments it is difficult to assess what is significant/realistic in the very detailed blow-by-blow accounts of these simulations. It would be much better if they could be presented in a much more condensed form, highlighting that main results, i.e., “cutting to the chase”. Many details could be relegated to appendices, supplementary material or archive.

This echoes the view of reviewer 1 that some of the results could be moved to supplementary material, and this has been done. Please see the response to reviewer 1. What we retained in the main text were the results we deemed most likely to appeal to an experimentalist.

(3) Personally I would question whether “it is possible to capture the rich properties of liquid foams by using a 2D model known as the viscous froth model”. On the face of it, this is claiming too much. The origin of the viscous effects that lie in at the heart of this model is the drag exerted by the two plates. There is no obvious counterpart in a typical 3d foam: viscous effects must arise otherwise and are likely to be very different in their effects. So I would recommend that less be claimed for the general significance of this work.

We agree with the reviewer: the original claim was overstated. We have changed the wording (in the introduction) from “to capture the rich properties of liquid foams” to “the properties of a foam layer flowing between two plates”. The point we were trying to make here is that the viscous froth model captures some of the rich properties of foam (but by no means does it capture all the complexity of 3D foam).

Appendix B

The authors have made considerable changes in response to my earlier comments, for which I thank them. The manuscript is now slightly shorter and more accessible. A highlight is the addition of the last paragraph of the conclusions, which outlines what an experimentalist might measure to verify these predictions, and the difficulties that might have to be overcome to perform the experiments. Nonetheless, this is still a complex piece of theoretical work that I think is worthy of publication in PRSA. I have only minor comments, and don't need see the manuscript again.

Page numbers refer to manuscript pages, rather than pages in the PDF file that I was sent.

1) In my opinion there are still spurious commas that disrupt the flow, e.g. p2 line 17 and line 48; pg 3, line 34; pg 8, line 42; pg 21, line 16; pg 25, line 38; and missing commas on pg 5, line 50; pg 13, line 41.

2) pg 3, line 20: it might be worth emphasising here that you mean only a straight channel (as the next paragraph makes clear, this is not true in a curved channel). I'm not sure that the statement "bubbles have the exact same shape no matter ... how fast they move" is true - I wondered if you mean that all bubbles have the same shape no matter how fast they move, even if that shape depends on speed, but later it is written that the bubbles move "without deforming". Doesn't this contradict the sort of structures found in ref 16?

Later in this paragraph, the authors have added the phrase "a bamboo foam was obtained experimentally"; I'm not sure what the point is here ... is it that this differs from theoretical predictions?

3) pg 3, line 26: the authors have updated some instances of dimensional quantities being referred to as small/large relative to the channel width, but I think this one, since it occurs early on, could also be changed. Then it might be worth being explicit at the top of page 9 that lengths are scaled by L and areas by L^2 . This would help with lines 50-51 on page 14.

4) pg 3, line 47: "beyond", rather than "after", a threshold?

5) pg 4, line 19: I'm not sure that anything was "proven" in [1], only demonstrated.

6) pg 4, fig 2: the authors might consider labelling the vertex, since it helps with the following paragraph.

7) pg 4, line 56: doesn't the use of "side wall" contradict the author's response letter concerning the use of upper and lower only? Shrinks to zero "length"?

- 8) pg 5, line 16: it might be useful to define equilibrium here, e.g. "(i.e. $p_b = 0$)"
- 9) pg 5, line 44: missing word after "appearing"
- 10) pg 5, line 51: missing "of" after "indication". Next line "transition" not "transitions"
- 11) pg 6, line 26: width -> channel?
- 12) pg 7, line 55: move "therefore" later in the sentence.
- 13) pg 8, lines 46-49: this sentence is really awkward to parse; I think it would help to re-write it.
- 14) pg 9, line 42: the superscript "o" notation was already used in section 2(a), so it might be better to introduce it there.
- 15) pg 10, line 10: details about what?
- 16) pg 10, line 14: the "imposed" back pressure, as we find out later, is not always imposed. This explains why it appears in the list of "unknown" variables, but this is confusing.
- 17) pg 9, lines 22-23; pg 10, lines 15-16; top of page 13; pg 14, line 44; pg 17, line 53; pg 25, line 9; are very repetitive on the subject of switching between coordinate systems. Where it is referred to on line 47 of pg 13, why not refer to the example that is in the manuscript (\S 3(a)(ii)) rather than, as currently, giving the impression that it is only used in the data shown in the SM?
- 18) pg 11, line 16: is the "total" turning angle one of the seven turning angles referred to earlier? If so, then "one of the total turning angles" might be better here.
- 19) pg 11, lines 29-30: I think this "meeting and annihilating" sentence could be reordered for clarity.
- 20) pg 10, line 46: with the new phrase added (in blue) i think "we will see later on does in fact" could be replaced with "can".
- 21) pg 18, first line: do we need to zoom into the zoomed region?
- 22) pg 23, line 9: is "exceedingly" necessary? Is this close in the Mr Kipling metric?
- 23) pg 25, line 34: I don't think you mean adding "three" bubbles - you have only added two so far.

Appendix C

- > Dear Editor of Proc Roy Soc A,
- > Thank you for sending the reviewer comments on our manuscript
- > RSPA-2021-0642.R1
- > We have noted that the reviewer has explicitly said
- > that they do not need to see the manuscript again.
- > However for the convenience of the editor, our detailed responses to
- > the reviewer are given below and also the latest round of changes to
- > the manuscript have been marked up in colour.
- > On the other hand, we also inform that we would like to opt for open access.
- > Yours sincerely,
- > Carlos Torres-Ulloa, Paul Grassia

The authors have made considerable changes in response to my earlier comments, for which I thank them. The manuscript is now slightly shorter and more accessible. A highlight is the addition of the last paragraph of the conclusions, which outlines what an experimentalist might measure to verify these predictions, and the difficulties that might have to be overcome to perform the experiments. Nonetheless, this is still a complex piece of theoretical work that I think is worthy of publication in PRSA. I have only minor comments, and don't need see the manuscript again.

Page numbers refer to manuscript pages, rather than pages in the PDF file that I was sent.

1) In my opinion there are still spurious commas that disrupt the flow, e.g. p2 line 17 and line 48; pg 3, line 34; pg 8, line 42; pg 21, line 16; pg 25, line 38; and missing commas on pg 5, line 50; pg 13, line 41.

- > Commas have been removed or inserted as instructed.

2) pg 3, line 20: it might be worth emphasising here that you mean only a straight channel (as the next paragraph makes clear, this is not true in a curved channel). I'm not sure that the statement "bubbles have the exact same shape no matter ... how fast they move" is true - I wondered if you mean that all bubbles have the same shape no matter how fast they move, even if that shape depends on speed, but later it is written that the bubbles move "without deforming". Doesn't this contradict the sort of structures found in ref 16?

- > The comment about the retaining the same shape referred specifically

> to the structures in Figure 1 in a straight channel, as has been
> clarified.

> There is no contradiction with ref [16] as that does not concern one
> of the shapes in Figure 1.

Later in this paragraph, the authors have added the phrase "a bamboo foam was obtained experimentally"; I'm not sure what the point is here ... is it that this differs from theoretical predictions?"

> We have expanded the discussion to explain that there is a maximum
> theoretical bubble area at which an infinite staircase must convert
> to a bamboo (although bamboos are permitted for areas smaller than
> this theoretical maximum).

> The reason we included the word "experimentally" was to highlight
> that experimentally the switch from staircase to bamboo happens at
> an area less than the theoretical maximum permitted area.

> We also made some minor modifications to the supplementary material
> to mention that the largest bubble areas in a three-bubble
> (equilibrium, but not necessarily monodisperse) staircase differ
> from largest bubble areas in a (monodisperse) infinite staircase

3) pg 3, line 26: the authors have updated some instances of dimensional quantities being referred to as small/large relative to the channel width, but I think this one, since it occurs early on, could also be changed. Then it might be worth being explicit at the top of page 9 that lengths are scaled by L and areas by L^2 . This would help with lines 50--51 on page 14.

> We have clarified that bubble size is relative to channel size.

> We have also explicitly mentioned that spatial coordinates are
> scaled by L , and areas are scaled by L^2 .

> We agree this helps with the text later on, since it is clearer what
> small (dimensionless) area and large (dimensionless) area means.

4) pg 3, line 47: "beyond", rather than "after", a threshold?

> The change has been made

5) pg 4, line 19: I'm not sure that anything was "proven" in [1], only demonstrated.

> Yes "demonstrated" is a better word: it is not a proof in the
> rigorous mathematical sense.

> Incidentally, prompted by this, for similar reasons we also changed
> another instance of the word "proven" from "proven to be
> unstable" to "found to be unstable"

6) pg 4, fig 2: the authors might consider labelling the vertex, since
it helps with the following paragraph.

> **Yes -- showing the vertex with an arrow and label in figure 2(a)
> is a good idea**

7) pg 4, line 56: doesn't the use of "side wall" contradict the
author's response letter concerning the use of upper and lower only?
Shrinks to zero "length"?

> "side wall" has been changed to "that wall" (so the
> contradiction with the earlier response letter is eliminated).

> "zero" has been changed to "zero length"

8) pg 5, line 16: it might be useful to define equilibrium here,
e.g. "(i.e. $p_b = 0$)"

> The definition has been added

9) pg 5, line 44: missing word after "appearing"

> Missing word has been added.

10) pg 5, line 51: missing "of" after "indication". Next line
"transition" not "transitions"

> Corrected.

11) pg 6, line 26: width -> channel?

> The sentence has been corrected.

12) pg 7, line 55: move "therefore" later in the sentence.

> Corrected.

13) pg 8, lines 46--49: this sentence is really awkward to parse; I think it would help to re-write it.

> The sentence has been split into two sentences and rewritten.

14) pg 9, line 42: the superscript "o" notation was already used in section 2(a), so it might be better to introduce it there.

> The superscript "o" notation was in fact already introduced prior
> to section 2(a), right at the start of section 2: we have
> highlighted the relevant text in green. We had merely reiterated the
> definition in section 2(c) (on page 9 as the reviewer mentioned). We
> changed section 2(c) to say "as mentioned earlier" to make it
> obvious that we were reiterating an earlier definition.

15) pg 10, line 10: details about what?

> Details of governing equations (as is now mentioned)

16) pg 10, line 14: the "imposed" back pressure, as we find out later, is not always imposed. This explains why it appears in the list of "unknown" variables, but this is confusing.

> The wording has been changed. We mention that we have to know values
> of 19 variables, but specifically no longer use the word "unknown"
> (since one of the 19 is indeed imposed, not unknown).

17) pg 9, lines 22--23; pg 10, lines 15--16; top of page 13; pg 14, line 44; pg 17, line 53; pg 25, line 9; are very repetitive on the subject of switching between coordinate systems. Where it is referred to on line 47 of pg 13, why not refer to the example that is in the manuscript (S 3(a)(ii)) rather than, as currently, giving the impression that it is only used in the data shown in the SM?

> It is correct that those "switched" cases are not considered
> solely in supplementary material (SM), but also in section 3(a)(ii).

> A comment to this effect has therefore been added as suggested.

18) pg 11, line 16: is the "total" turning angle one of the seven turning angles referred to earlier? If so, then "one of the total turning angles" might be better here.

> This change has been made

19) pg 11, lines 29-30: I think this "meeting and annihilating" sentence could be reordered for clarity.

> The sentence has been split into two, which we agree is clearer.

20) pg 10, line 46: with the new phrase added (in blue) I think "we will see later on does in fact" could be replaced with "can".

> This replacement has been made (it was actually page 11, line 46, in
> the previous manuscript, not page 10, line 46 as the reviewer
> stated)

21) pg 18, first line: do we need to zoom into the zoomed region?

> Reworded: we now say "in the zoomed view" rather than "zooming
> into (the zoomed view)"

22) pg 23, line 9: is "exceedingly" necessary? Is this close in the Mr Kipling metric?

> "exceedingly close to" has been changed to "close to"

23) pg 25, line 34: I don't think you mean adding "three" bubbles - you have only added two so far

> "adding" has been changed to "having"